# RobustFlow: Towards Robust Agentic Workflow Generation

## Abstract

The automated generation of agentic workflows is a promising frontier for enabling large language models (LLMs) to solve complex tasks. However, our investigation reveals that the robustness of agentic workflow remains a critical, unaddressed challenge. Current methods often generate wildly inconsistent workflows when provided with instructions that are semantically identical but differently phrased. This brittleness severely undermines their reliability and trustworthiness for real-world applications. To quantitatively diagnose this instability, we propose metrics based on nodal and topological similarity to evaluate workflow consistency against common semantic variations such as paraphrasing and noise injection. Subsequently, we further propose a novel training framework, RobustFlow, that leverages preference optimization to teach models invariance to instruction variations. By training on sets of synonymous task descriptions, RobustFlow boosts workflow robustness scores to 70% - 90%, which is a substantial improvement over existing approaches. The code is publicly available at https://anonymous.4open.science/r/RobustFlow-1B19.

## 1 Introduction

The paradigm of employing Large Language Models (LLMs) to construct and orchestrate agentic workflows has emerged as a highly promising direction for tackling complex, multi-step tasks. By decomposing intricate problems into a structured sequence of actions, these workflows significantly enhance the capabilities of LLMs (Khattab et al., 2024; Tang et al., 2023). Consequently, the automated generation of such workflows has become a critical area of research, aiming to supplant the laborious and expertise-driven process of manual construction (Zhang et al., 2024). Recent efforts in this domain have led to methods that can either learn a general-purpose workflow for a specific task domain (task-level) (Zhang et al., 2024; Hu et al., 2024; Zhang et al., 2025a) or dynamically generate a bespoke workflow for each query (query-level) (Gao et al., 2025; Wang et al., 2025; Niu et al., 2025). A growing body of evidence suggests that these automatically generated workflows can achieve superior effectiveness and generalization compared to their manually designed counterparts (Yuksekgonul et al., 2024; Shang et al., 2024; Li et al., 2024).

However, for any agentic system to be genuinely dependable, high performance on its own is inadequate. A critical but overlooked attribute is robustness, which is the capacity to yield consistent and stable outputs when faced with semantically similar input variations. (Wang et al., 2020; Zhu et al., 2023b; Pei et al., 2024). Our investigation reveals that current state-of-the-art agentic workflow methods are deficient in robustness. When provided with semantically equal but differently phrased queries, the workflows they produce can only retain 70% to even 40% stability. Such inconsistency is a major barrier to their credibility and practical deployment. It is particularly telling that, as confirmed by prior research (Song et al., 2024; He & Lab, 2025; Atil et al., 2024), this instability remains even when the sampling temperature of LLM is reduced to zero, suggesting the problem is not a simple artifact of randomness but a deeper failure to achieve semantic invariance.

In this paper, we conduct the first quantitative analysis of robustness in automated agentic workflow generation. To enable this, we constructed a comprehensive dataset designed to test model stability against query variations. We began by collecting 1,255 base task descriptions across 6 diverse tasks and, for each one, systematically generated a suite of 6 synonymous but formally distinct variations using techniques like requirement augmentation, paraphrasing, and noise injection. This process

Table 1: Summary of automatic agentic workflow generation methods.

| Method | Level | Task | Edge | Code Idea |
|--------|-------|------|------|-----------|
| ADAS | Task | QA/MATH/Reasoning | Code | Meta Agent Search |
| AFlow | Task | QA/Code/MATH | Code | MCTS search |
| MaAS | Task | Code/MATH | Graph | Agentic Supernet |
| FlowReasoner | Query | QA/Code/MATH | Code | Reasoning RL |
| ScoreFlow | Query | QA/Code/MATH | Code | Score-based DPO |
| Flow | Query | Code | Graph | Modularity & Updating |
| RobustFlow | Query | QA/Code/MATH | Code | Robust Generation |

yielded a testbed of 31,889 workflows generated from these description variations. By evaluating the structural and semantic integrity of these workflows against reference standards, our analysis reveals a critical lack of robustness across existing methods, demonstrating that their stability varies significantly with both the generation approach and task complexity.

To address such robustness limitations, we propose RobustFlow, a novel framework that trains the model to generate a single, canonical workflow based on synonymous queries using preference optimization. During training, we automatically generate semantically equivalent queries to prepare a batch of workflows for training. From this batch, the most frequent and effective structure is designated as the positive example, while structurally divergent ones are treated as negative examples for the preference-based training. Our contributions are threefold:

- We identify and investigate the critical problem of robustness in agentic workflow generation. Our empirical analysis of existing methods reveals the general existence of this problem and key insights into how stability is affected by different generation strategies and task types.

- We introduce an evaluation suite to quantitatively analyze the problem: a novel methodology for measuring workflow robustness based on nodal consistency and topological similarity; and a new dataset comprises 31,889 workflows generated from 1,255 task descriptions across 6 domains, each systematically altered with 5 types of semantic variations.

- We introduce RobustFlow, a training framework that directly addresses the identified robustness issues. By leveraging preference optimization, RobustFlow significantly improves the stability of agentic workflows, boosting robustness scores to 70% - 90%.

## 2 RELATED WORK

**Agentic Workflows in Multi-Agent Systems.** LLM agents are increasingly composed into multi-agent systems (MAS), enabling long-horizon planning and collaboration, which in turn improves performance on complex real-world tasks (Wang et al., 2024; Hong et al., 2024b; Chen et al., 2023a; Zhu et al., 2023a). However, many MAS still rely on manually crafted workflows and rules, which generalize poorly to unseen tasks and open-ended settings (Chen et al., 2023b; Qian et al., 2023). This gap naturally motivates the automated design of agentic workflows.

Formally, given an input space $\mathcal{Q}$ and a set of callable agents $\mathcal{A}$, an automated workflow generator $G_\theta : \mathcal{Q} \times \mathcal{A} \to \mathcal{W}$ maps a user query $q \in \mathcal{Q}$ to an executable workflow $w = G_\theta(q; \mathcal{A}) \in \mathcal{W}$, where $\mathcal{W}$ denotes the search space of executable workflows and $\theta$ denotes the trainable parameters of the generator. For any workflow $w$, we define its normalized graph representation as $\Gamma(w) = (V, E)$. Each node $v \in V$ denotes a single agent invocation, and each directed edge $e \in E$ jointly encodes data and control dependencies. At the implementation level, workflows can be represented either as graphs or as code. Based on the input granularity of $q$, existing approaches can be classified as task-level and query-level. Representative methods are summarized in Table 1.

Task-level methods build a single workflow $w$ for each task family $\{q^{(i)}\}_{i=1}^n \subseteq \mathcal{Q}$. ADAS (Hu et al., 2024) unifies agentic workflows in code representation and employs meta-agent search for automatic construction and optimization. AFlow (Zhang et al., 2024) represents workflows as operator graphs and uses MCTS to explore the structural space for efficient pipelines. MaAS (Zhang et al., 2025a) adopts an agentic supernet to trade off performance and cost via distributions over architectures.

Query-level methods generate a customized workflow $w$ for a single user query $q \in \mathcal{Q}$. FlowReasoner (Gao et al., 2025) leverages an O1-like reasoning meta-agent to synthesize workflows without large-scale search. ScoreFlow (Wang et al., 2025) trains a workflow generator with DPO for higher-quality per-query plans. Flow (Niu et al., 2025) models workflows as activity-on-vertex graphs (AOV) and reallocates subtasks on the fly using historical performance and prior AOVs.

**Robustness of LLMs.** LLMs often generate noticeably inconsistent responses to semantically equivalent instructions that are phrased differently (Wang et al., 2020; Zhu et al., 2023b). This reveals the problem of insufficient robustness of LLMs (Moradi & Samwald, 2021; Chao et al., 2024), which undermines the reliability and controllability in practice. Existing research generally alleviates the problem from two complementary directions: the training phase and the inference phase, both aimed at narrowing the gap between task semantics and responses.

In the training phase, Adversarial training (Goodfellow et al., 2014) applies gradient-based worst-case perturbations to each instruction. Consistency regularization (Kou et al., 2024) adds a Consistency Loss on top of language modeling to minimize the KL divergence between the predicted distributions of semantically equivalent inputs. Contrastive instruction tuning (Yan et al., 2024) uses a contrastive loss to pull representations of equivalent expressions closer and push non-equivalent ones apart. Distributionally robust optimization (Zhao et al., 2024; Fisch et al., 2024) combines preference pairs with self-supervised contrasts to minimize consistency risk across expression variants. In the inference phase, pretrained rewriter (Fu et al., 2024) uses a lightweight and pretrained module to normalize user instructions into the model-preferred phrasing. Multi-view voting (Chen et al., 2024; Yao, 2024) generates several paraphrases for each instruction and aggregates the outputs by majority vote or confidence-weighted fusion.

## 3 PRELIMINARY

We formulate the robust agentic workflow generation problem in Section 3.1, define the input perturbation protocol in Section 3.2, and present structure-aware robustness metrics in Section 3.3.

### 3.1 PROBLEM DEFINITION

To characterize robustness under input perturbations, our perturbation protocol (details in Sec. 3.2) specifies a family of intensity-indexed distributions $\{\mathcal{P}_k\}$, where $k$ represents disturbance intensity. For any perturbation $\delta \sim \mathcal{P}_k$, we define the perturbed input $q + \delta$ and corresponding workflow $w_\delta = G_\theta(q + \delta; \mathcal{A})$. To measure structural stability, we adopt a structure-sensitive similarity metric $\mathcal{F} : \mathcal{W} \times \mathcal{W} \to [0, 1]$, which satisfies $\mathcal{F}(w, w) = 1$ for identical workflows. We detail the specific computation of $\mathcal{F}$ based on nodal and topological consistency in Sec. 3.3. Accordingly, we define the robustness risk $R_k(G_\theta)$ of a workflow generator $G_\theta$ as the expected structural discrepancy between the perturbed workflow $G_\theta(q + \delta; \mathcal{A})$ and the original workflow $G_\theta(q; \mathcal{A})$:

$$R_k(G_\theta) \;=\; \mathbb{E}_{q \sim \mathcal{Q}, \, \delta \sim \mathcal{P}_k} \Big[ 1 - \mathcal{F}\big(G_\theta(q; \mathcal{A}), \, G_\theta(q + \delta; \mathcal{A})\big) \Big]. \tag{1}$$

Our objective is to find a workflow generator $G_\theta \in \mathcal{G}$ that minimizes the risk $R_k(G_\theta)$:

$$G_\theta^\star \;=\; \arg \min_{G_\theta \in \mathcal{G}} R_k(G_\theta). \tag{2}$$

**Workflow Representation.** To ensure comparability and implementation-agnostic evaluation, we map code-represented workflows into a unified graph structure form, with detailed examples provided in Appendix A.2. Sequences and branches are mapped to directed edges capturing data and control dependencies. Loops are unrolled into finite replicas of functional nodes, whose outputs are subsequently processed by downstream nodes. Through this normalization, the top-level representation $\Gamma(w)$ is formulated as a directed acyclic graph (DAG) (Kahn, 1962). We evaluate the robustness similarity $\mathcal{F}(w_1, w_2)$, along with all subsequent structural comparisons, on the normalized DAG, and we detail the definitions and procedures in Sec. 3.3.

### 3.2 PERTURBATION PROTOCOL

For each original task $q \in \mathcal{Q}$, we define its semantic cluster $C(q)$ as the set comprising the original task and its $N$ generated variants:

$$\mathcal{C}(q) \;=\; \{q\} \cup \{q^{(i)} \mid q^{(i)} \equiv q, \, i = 1, \ldots, N\}, \tag{3}$$

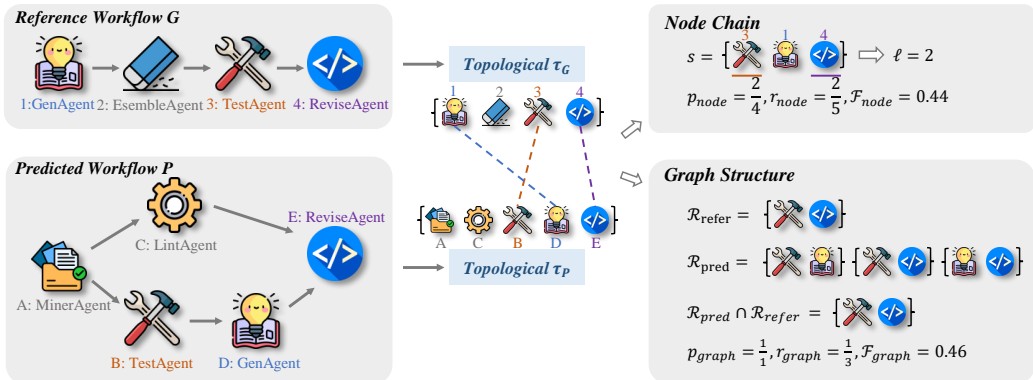

Figure 1: **Structure-aware robustness evaluation metrics.** We first align nodes between the reference and predicted workflows to establish a correspondence. Based on this alignment, we compute two key metrics, node-chain robustness derived from the longest increasing subsequence length $l$ of the topological order, and graph-structure robustness calculated by comparing the preservation of reachability dependencies on the aligned DAGs.

where $\equiv$ denotes semantic equivalence, implying that $q^{(i)}$ shares the identical task objective with $q$ despite differing in phrasing or specific constraints.

Following below perturbation protocol, for each task, we sample an intensity level $k$ and a specific perturbation operation $\delta \sim \mathcal{P}_k$ to construct a variant $q^{(i)} = q + \delta$. All subsequent evaluations are performed within $\mathcal{C}(q)$, comparing workflows induced by these variants. To systematically characterize input-side variations, we introduce three categories of perturbations (more detailed prompts are provided in Fig. 9 and Appendix C):

- **Paraphrasing**. We paraphrase instructions using LLMs, modifying tense, voice, sentence structure, and wording while preserving semantics and all task constraints.

- **Requirement Augmentation**. Without altering the task objective, we add or tighten executable constraints via LLMs, such as limits on steps or time, or output format templates. We prefer feasibility-preserving constraints so that at least one valid workflow is guaranteed to exist.

- **Noise injection**. Following the TextAttack framework (Morris et al., 2020), we apply random synonym substitution, insertion, swap, and deletion at the word level. Based on the perturbation rate $\rho$ (percentage of modified tokens), we categorize noise intensity into light ($\rho \in [0.2, 0.4]$), moderate ($\rho \in [0.4, 0.6]$), and heavy ($\rho \in [0.6, 0.8]$). To prevent semantic collapse, we mask protected spans from random edits, including numbers, variable names, proper nouns, and mathematical formulas.

To ensure data quality and reproducibility, we conducted a manual verification of all 7,530 rewritten variants (involving 5 authors over 2 months). We retain only variants that are semantically aligned with the original task, preserve non-contradictory and feasible constraints, and maintain valid formatting such as code blocks, math markup. The final robustness evaluation dataset comprises 1,255 semantic clusters, yielding a total of 7,530 validated instruction variants.

## 3.3 STRUCTURE-AWARE ROBUSTNESS EVALUATION

To ensure reliable evaluation, we quantify robustness with two complementary structure-aware measures: node-level similarity over a topological sequence and graph-level structural similarity on the DAG, as shown in Fig. 1. For a semantic cluster $\mathcal{C}$, we take the workflow induced by the original formulation as the reference workflow $w^g$ with nodes $V^g$ and edges $E^g$. For any other formulation $q^{(i)} \in \mathcal{C}$, we denote its workflow as the predicted workflow $w^p$ with nodes $V^p$ and edges $E^p$.

**Node Alignment.** We first align semantic steps between the two workflows based on their textual content. We compute a pairwise similarity matrix $S \in \mathbb{R}^{|V^g| \times |V^p|}$ where each entry represents the cosine similarity between node embeddings. To filter out weak associations, we apply a pruning

threshold $\beta$ and define the sparse similarity matrix $S$ as:

$$S_{i,j} = \begin{cases} \sigma(v_i^g, v_j^p), & \text{if } \sigma(v_i^g, v_j^p) \geq \beta, \\ 0, & \text{otherwise.} \end{cases} \quad (4)$$

Treating the nonzero entries of $S$ as weighted edges in a bipartite graph, we solve the maximum-weight bipartite matching problem. This yields the optimally matched subsets of nodes $V^{g'} \subseteq V^g$ and $V^{p'} \subseteq V^p$, along with a one-to-one mapping $\pi : V^{p'} \to V^{g'}$ that links each matched predicted node to its semantic counterpart in the reference workflow.

**Node Chain.** We measure node chain robustness by the degree to which the relative order of matching nodes in the prediction chain is preserved. Following the T-eval (Chen et al., 2023c) and Worf-Bench (Qiao et al., 2024), let $\tau_p$ be a topological sequence of the predicted DAG restricted to $V^{p'}$. Mapping each $v \in V^{p'}$ to its counterpart via $\pi$ and then to its index under a reference topological sequence $\tau_g$, we obtain an index sequence $\mathbf{s} = \left[\text{idx}_{\tau_g}(\pi(v))\right]_{v \in \tau_p}$.

The robustness of relative ordering is measured by the length $l$ of the longest increasing subsequence (LIS) of $\mathbf{s}$. We define robustness score of node chain $\mathcal{F}_{\text{node}}$ as:

$$p_{\text{node}} = \frac{l}{|V^p|}, \qquad r_{\text{node}} = \frac{l}{|V^g|}, \qquad \mathcal{F}_{\text{node}} = \frac{2\,p_{\text{node}}\,r_{\text{node}}}{p_{\text{node}} + r_{\text{node}}}, \quad (5)$$

where $p_{\text{node}}$ and $r_{\text{node}}$ represent the precision and recall of the generated node chain, respectively. When multiple reference topological sequences exist, we select the one maximizing $l$.

**Graph Structure.** We measure graph structure robustness by whether the predicted workflow correctly preserves the dependency structure between tasks. We project predicted edges onto reference nodes via the alignment. For each predicted edge $(u^p, v^p)$ with $u^p, v^p \in V^{p'}$, add $(\pi(u^p), \pi(v^p))$ to $E^{p' \to g'}$. On the common node set $V^{g'}$, define reachability pairs as:

$$\mathcal{R}_{\text{pred}} = \{(u,v) \mid u \rightsquigarrow v \text{ in } (V^{g'}, E^{p' \to g'})\}, \quad \mathcal{R}_{\text{refer}} = \{(u,v) \mid u \rightsquigarrow v \text{ in } (V^{g'}, E^{g'})\}, \quad (6)$$

where $\rightsquigarrow$ denotes the existence of a directed path from $u$ to $v$. Based on the reachability pair sets, we define the robustness score of graph structure $\mathcal{F}_{\text{graph}}$ as:

$$p_{\text{graph}} = \frac{|\mathcal{R}_{\text{pred}} \cap \mathcal{R}_{\text{refer}}|}{|\mathcal{R}_{\text{pred}}|}, \quad r_{\text{graph}} = \frac{|\mathcal{R}_{\text{pred}} \cap \mathcal{R}_{\text{refer}}|}{|\mathcal{R}_{\text{refer}}|}, \quad \mathcal{F}_{\text{graph}} = \frac{2p_{\text{graph}}r_{\text{graph}}}{p_{\text{graph}} + r_{\text{graph}}}, \quad (7)$$

where $p_{\text{graph}}$ and $r_{\text{graph}}$ represent the precision and recall of the generated graph structure.

# 4 ROBUSTFLOW

While existing methods advance performance, efficiency, and search strategies (Jaggavarapu, 2025; Zhang et al., 2025b; Trirat et al., 2025), they devote limited attention to structural robustness in realistic deployments. Under perturbations, the induced workflow structures can vary substantially, undermining stability and reliability in practice. To address this gap, we build RobustFlow, a robustness-oriented agentic workflow generation method trained via a two-stage training pipeline. As shown in Fig. 2, RobustFlow first performs instruction-augmented supervised fine-tuning to mitigate the cold-start, then applies self-consistency preference optimization to enhance structural robustness and consistency. The two-stage pipeline grounds our design and clarifies how dataset curation and modeling choices jointly improve robustness to input perturbations.

## 4.1 INSTRUCTION AUGMENTED SUPERVISED FINE-TUNING

To mitigate cold-start issues in Reinforcement Learning and instill initial structural invariance, we first perform instruction-augmented supervised fine-tuning. This enables the model to learn the mapping from diverse instructions to executable workflows and the associated structural constraints.

**Instruction Augmentation.** To equip the model with robust prior knowledge against input perturbations, we rewrite only the instruction and keep the corresponding workflow unchanged within each cluster. We use FLORA-Bench (Zhang et al., 2025b) as the base instruction-workflow dataset,

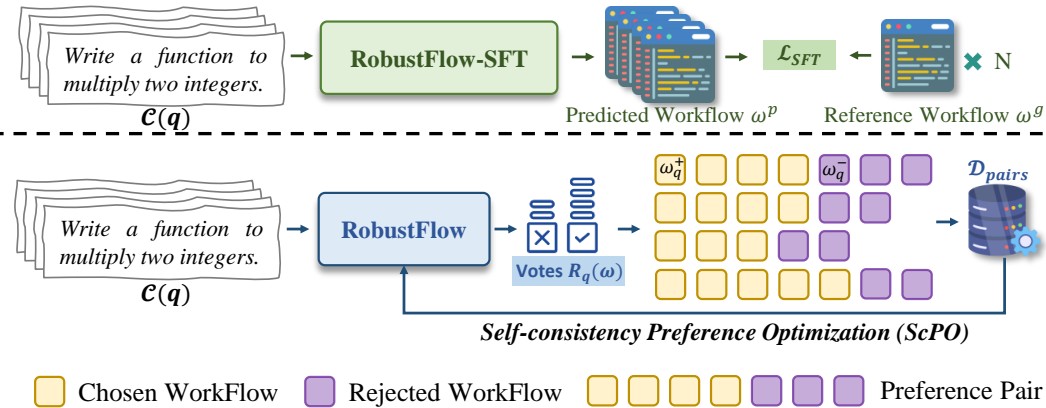

Chosen WorkFlow ☐   Rejected WorkFlow ☐   ☐☐☐☐ ☐ ☐ ☐ Preference Pair

Figure 2: **Overview of RobustFlow.** RobustFlow first performs instruction-augmented supervised fine-tuning to mitigate the cold-start, then applies self-consistency preference optimization to enhance structural robustness and consistency.

denoted by $\mathcal{D}_0 = \{(q_n, w_n^g)\}_{n=1}^{N_0}$, where $q_n$ is the original instruction and $w_n^g$ is the reference workflow. Following the perturbation protocol defined in Sec. 3.2, for each $q_n$, we generate semantically preserving variants $q_n^{(i)} = q_n + \delta^{(i)}$ (where $i = 0$ denotes the original) and construct a semantic cluster $\mathcal{C}(q_n)$. We define the instruction-augmented SFT dataset $\mathcal{D}_{\text{SFT}}$ as:

$$\mathcal{D}_{\text{SFT}} = \bigcup_{n=1}^{N_0} \left\{ \left( q_n^{(i)}, w_n^g \right) \mid i = 0, \ldots, K \right\}, \tag{8}$$

where $N_0$ denotes the number of semantic clusters, with $K$ denoting the variant count per cluster.

**Supervised Fine-Tuning.** We fine-tune the generator on the augmented instruction dataset $\mathcal{D}_{\text{SFT}}$. Treating the ground-truth workflow $w^g$ as a token sequence $y = (y_1, \ldots, y_T)$, we optimize the standard supervised next-token cross-entropy objective:

$$\mathcal{L}_{\text{SFT}} = \mathbb{E}_{(q,y) \sim \mathcal{D}_{\text{SFT}}} \left[ -\sum_{t=1}^{T} \log P_\theta \left( y_t \mid q, \ y_{<t} \right) \right], \tag{9}$$

where $y_t$ denotes the $t$-th token, and $y_{<t}$ denotes its prefix. This stage equips the generator with fundamental workflow generation capability and basic structural invariance, thereby laying a solid foundation for the subsequent self-consistency preference optimization.

## 4.2 SELF-CONSISTENCY PREFERENCE OPTIMIZATION

Starting from the SFT-augmented model $M_0$, we refine the generator via an iterative cluster-aware self-consistency preference optimization (ScPO). The key idea is to mine preference pairs $(w^+, w^-)$ within each semantic cluster $\mathcal{C}(q)$ by combining execution scores with self-consistency votes.

At iteration $t$, using the current generator model $M_t$, we sample $r$ candidate workflows for each query variant in cluster $\mathcal{C}(q)$ and aggregate them into a candidate set $\mathcal{Y}_q$. Each candidate $w$ is canonicalized to its normalized DAG $\Gamma(w)$. Let $\text{uniq}(\mathcal{Y}_q)$ be the set of unique workflows after canonicalization. To characterize preferences, we define the execution score $s_q(w) \in [0, 1]$ by directly adopting the standard evaluation metric for each task, avoiding the need for an additional learned reward model. Specifically, $s_q(w)$ is instantiated as the final-answer accuracy for math benchmarks (GSM8K, MATH), the unit test pass rate for code generation benchmarks (MBPP, HumanEval), and the F1 score for question answering benchmarks (HotpotQA, DROP). Additionally, we define the self-consistency vote count $v_q(w)$ as:

$$v_q(w) = \left| \{ w' \in \mathcal{Y}_q : \Gamma(w') = \Gamma(w) \} \right|. \tag{10}$$

Here, $s_q(w)$ measures functional correctness, whereas $v_q(w)$ reflects structural frequency. To implement a "score-first, vote-second" aggregation, we define the preference score $R_q(w)$ as:

$$R_q(w) = s_q(w) + \lambda \frac{v_q(w)}{|\mathcal{Y}_q|}, \tag{11}$$

Table 2: Comparison of robustness performance among agentic workflow generation methods on Code, Math, and QA benchmarks under five perturbation types. We evaluate both node-level and graph-level robustness and report the average over ten independent runs.

| Perturbation | Method | Code | | | | MATH | | | | QA | | | | Avg. |
|---|---|---|---|---|---|---|---|---|---|---|---|---|---|---|
| | | MBPP | | HumanEval | | GSM8K | | MATH | | HotpotQA | | DROP | | |
| | | Node | Graph | Node | Graph | Node | Graph | Node | Graph | Node | Graph | Node | Graph | |
| Requirement | AFlow | 0.57 | 0.79 | 0.62 | 0.47 | 0.37 | 0.27 | 0.31 | 0.11 | 0.56 | 0.50 | 0.56 | 0.51 | 0.44 |
| | Flow | **0.91** | **0.83** | 0.73 | 0.42 | - | - | - | - | - | - | - | - | 0.63 |
| | ScoreFlow | 0.77 | 0.75 | 0.89 | 0.86 | 0.71 | 0.69 | 0.77 | 0.60 | 0.66 | 0.72 | 0.66 | 0.63 | 0.71 |
| | RobustFlow | 0.89 | **0.83** | **0.95** | **0.93** | **0.87** | **0.77** | **0.84** | **0.80** | **0.88** | **0.83** | **0.86** | **0.82** | **0.82** |
| Paraphrasing | AFlow | 0.51 | 0.75 | 0.76 | 0.61 | 0.41 | 0.34 | 0.59 | 0.42 | 0.55 | 0.29 | 0.51 | 0.52 | 0.49 |
| | Flow | 0.82 | 0.73 | 0.78 | 0.67 | - | - | - | - | - | - | - | - | 0.70 |
| | ScoreFlow | 0.84 | 0.73 | 0.94 | 0.92 | 0.66 | 0.67 | 0.83 | 0.76 | 0.64 | 0.64 | 0.63 | 0.66 | 0.73 |
| | RobustFlow | **0.92** | **0.88** | **0.98** | **0.96** | **0.91** | **0.88** | **0.87** | **0.83** | **0.93** | **0.88** | **0.94** | **0.85** | **0.88** |
| Light Noise | AFlow | 0.58 | 0.67 | 0.60 | 0.52 | 0.42 | 0.36 | 0.33 | 0.16 | 0.51 | 0.34 | 0.52 | 0.49 | 0.42 |
| | Flow | 0.81 | 0.83 | 0.67 | 0.45 | - | - | - | - | - | - | - | - | 0.64 |
| | ScoreFlow | 0.54 | 0.71 | 0.87 | 0.71 | 0.64 | 0.53 | 0.71 | 0.51 | 0.67 | 0.57 | 0.71 | 0.57 | 0.60 |
| | RobustFlow | **0.95** | **0.93** | **0.96** | **0.90** | **0.92** | **0.85** | **0.88** | **0.89** | **0.91** | **0.87** | **0.94** | **0.89** | **0.89** |
| Moderate Noise | AFlow | 0.64 | **0.79** | 0.73 | 0.64 | 0.48 | 0.45 | 0.41 | 0.28 | 0.64 | 0.48 | 0.58 | 0.57 | 0.54 |
| | Flow | 0.69 | 0.71 | 0.53 | 0.59 | - | - | - | - | - | - | - | - | 0.65 |
| | ScoreFlow | 0.63 | 0.67 | 0.73 | 0.65 | 0.61 | 0.51 | 0.60 | 0.53 | 0.44 | 0.33 | 0.63 | 0.47 | 0.53 |
| | RobustFlow | **0.82** | 0.78 | **0.87** | **0.84** | **0.78** | **0.71** | **0.83** | **0.75** | **0.85** | **0.82** | **0.83** | **0.79** | **0.78** |
| Heavy Noise | AFlow | **0.75** | **0.84** | 0.78 | 0.71 | 0.61 | 0.68 | 0.63 | 0.44 | 0.69 | 0.59 | 0.50 | 0.64 | 0.65 |
| | Flow | 0.64 | 0.40 | 0.42 | 0.33 | - | - | - | - | - | - | - | - | 0.37 |
| | ScoreFlow | 0.48 | 0.46 | 0.81 | 0.68 | 0.61 | 0.53 | 0.49 | 0.45 | 0.37 | 0.49 | 0.58 | 0.34 | 0.49 |
| | RobustFlow | 0.72 | 0.69 | **0.90** | **0.82** | **0.68** | **0.72** | **0.78** | **0.66** | **0.78** | **0.75** | **0.78** | **0.71** | **0.72** |

where $\lambda$ is a coefficient ensuring the precedence of execution quality. In practice, we enforce this strictly via lexicographical sorting (primary key: $s_q$, secondary key: $v_q$), which is mathematically equivalent to setting a sufficiently small $\lambda$. We then select the extremal pair:

$$w_q^+ = \arg \max_{w \in \text{uniq}(\mathcal{Y}_q)} R_q(w), \ w_q^- = \arg \min_{w \in \text{uniq}(\mathcal{Y}_q)} R_q(w), \tag{12}$$

to form the preference-pair dataset $\mathcal{D}_{\text{pairs}}$. We update the model $M_\theta$ (initialized from $M_t$) by minimizing the following weighted objective:

$$\mathcal{L}_{\text{ScPO}}(q) = -\rho_q \log \sigma \Big( \beta \log \frac{M_\theta(w_q^+ \mid q)}{M_t(w_q^+ \mid q)} - \beta \log \frac{M_\theta(w_q^- \mid q)}{M_t(w_q^- \mid q)} \Big) - \alpha \, \rho_q \frac{1}{|w_q^+|} \log M_\theta(w_q^+ \mid q), \tag{13}$$

Here, the confidence weight $\rho_q = R_q(w_q^+) - R_q(w_q^-)$ scales the gradient so that the model learns more from pairs with a distinct quality gap. The second term is a negative log-likelihood (NLL) regularizer weighted by $\alpha$, which stabilizes training and prevents the probability of high-quality samples from vanishing. After optimization, we update $M_{t+1} \leftarrow M_\theta$ and proceed to the next iteration of sampling and training.

## 5 EXPERIMENTS

### 5.1 EXPERIMENTAL SETUP

**Datasets.** Following prior practice of AFlow (Zhang et al., 2024), we collect 1,255 original task descriptions from six public benchmarks spanning three task domains: math reasoning, question answering, and code generation. We then construct five perturbation variants for each original description, which form 1,255 semantic clusters together with the originals. Within each cluster, we perform repeated sampling and generation, yielding 7,530 instruction variants and 31,889 corresponding workflows. More details are available in the Appendix A.3.

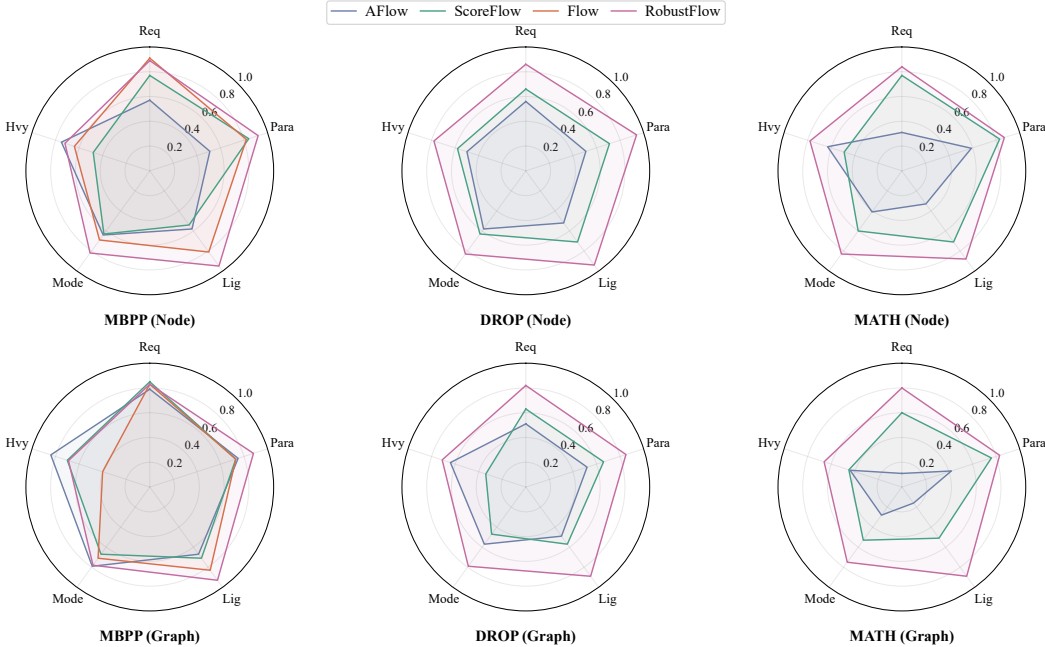

Figure 3: Robustness of agentic workflow generation methods under perturbations on MBPP, DROP, and MATH. Colors in the legend denote methods. Dimensions: Req = Requirement Augmentation, Para = Paraphrasing, Lig/Mode/Hvy = Light/Moderate/Heavy noise.

**Baselines.** We compare the performance of workflows generated by six agentic workflow generation methods, including AutoAgents (Chen et al., 2023a), ADAS (Hu et al., 2024), AFlow (Zhang et al., 2024), MaAS (Zhang et al., 2025a), ScoreFlow (Wang et al., 2025), and FlowReasoner (Gao et al., 2025). For the robustness evaluation, we only compare the fully open-source ones (AFlow, Flow, and ScoreFlow) as they require executing the full pipelines under controlled perturbations.

Table 3: Comparison of workflow performance for automated agentic workflow generation in Code scenarios. Each method is executed with GPT-4o-mini, and we repeat experiments three times for the average score.

| Method | HumanEval | MBPP | Avg. |
|---|---|---|---|
| AutoAgents (Chen et al., 2023a) | 88.91 | 72.03 | 80.47 |
| ADAS (Hu et al., 2024) | 84.26 | 68.47 | 76.37 |
| AFlow (Zhang et al., 2024) | 94.15 | 82.40 | 88.28 |
| MaAS (Zhang et al., 2025a) | 95.42 | 84.16 | 89.80 |
| ScoreFlow (Wang et al., 2025) | 95.90 | 84.70 | 90.30 |
| FlowReasoner (Gao et al., 2025) | **97.26** | **92.15** | **94.71** |
| RobustFlow | 93.67 | 81.90 | 87.79 |

**Implementation Details.** By default, RobustFlow utilizes Qwen3-32B (Yang et al., 2025) as the base model for the generator (inference via vLLM (Kwon et al., 2023)) and GPT-4o-mini (Hurst et al., 2024) as the executor (inference via API). The temperature for all models is set to 0. All experiments are carried out on servers equipped with 8 NVIDIA H100 80GB GPUs, and we fine-tune with LoRA (Hu et al., 2022) using the ms-swift (Zhao et al., 2025) framework. Further implementation details are provided in Appendix A.4.

**Metrics.** In the robustness evaluation experiments, we use the $\mathcal{F}_{\text{node}}$ and $\mathcal{F}_{\text{graph}}$ (Sec . 3.3) as the primary metrics. In the performance evaluation experiments, we report the pass@1 metric, as presented in (Chen et al., 2021), to assess code accuracy.

## 5.2 RESULTS AND ANALYSIS

**Robustness Evaluation.** Table 2, Figs. 3 and 7 illustrate that existing workflow generation methods generally suffer from severe robustness issues when faced with perturbations. Task-level methods are particularly vulnerable, with workflow stability dropping to approximately 40% under moderate

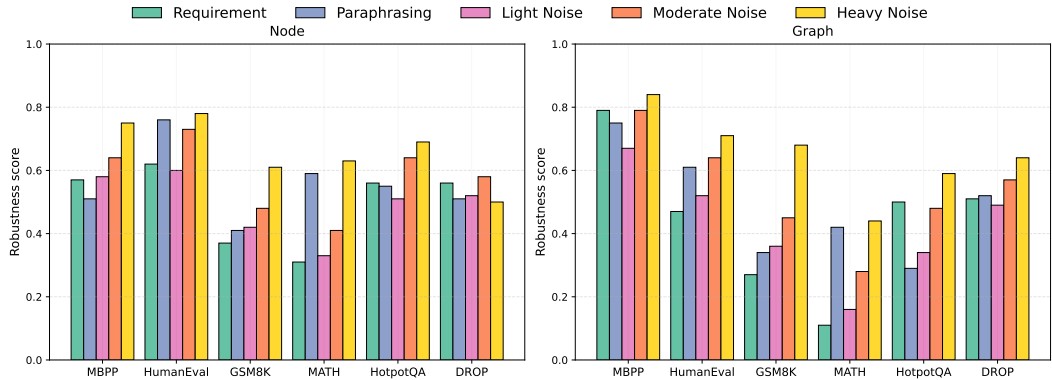

Figure 5: Dataset sensitivity analysis across three domains (Code > QA > Math). Structured code tasks exhibit higher inherent stability compared to linguistically nuanced QA tasks.

descriptive changes. In contrast, RobustFlow consistently achieves balanced and superior robustness across all perturbation types and datasets.

Fig. 4 further reports noise-level trends averaged across datasets. Task-level methods such as AFlow are particularly fragile under paraphrasing and light-moderate noise, yet show a counterintuitive increase when noise becomes heavy. Query-level methods peak under paraphrasing and degrade steadily as noise intensifies, even when the sampling temperature of LLMs is reduced to zero, suggesting the problem is not a simple artifact of randomness. RobustFlow remains the best robustness across all levels, while still retaining query-level characteristics with only a slight decrease from light to heavy noise.

We attribute the exceptional rise of task-level methods under heavy noise to input insensitivity. When instructions are heavily disrupted, these methods fail to parse specific intention and instead revert to a static general task backbone, resulting in artificially high consistency scores. In contrast, query-level methods like RobustFlow attempt to reconstruct specific plans from remaining keywords. Their performance decreases accordingly as semantic information is lost, reflecting a necessary trade-off between customizability and noise tolerance.

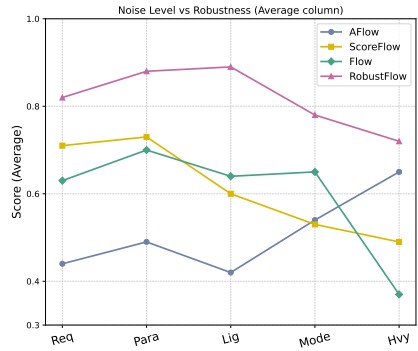

Figure 4: Robustness trends of different methods under noise enhancement.

**Performance Comparison.** Table 3 compares the performance of workflow generation methods on code benchmarks. RobustFlow achieves an average of 87.79, which is slightly lower than the best-performing query-level methods such as FlowReasoner (94.71) and ScoreFlow (90.30). This suggests that robustness-oriented optimization incurs a modest trade-off in raw performance, but the reduction remains within an acceptable range.

This phenomenon reflects a deliberate shift in optimization objectives. Existing methods essentially act as experts, maximizing rewards for specific prompts, but often rely on fragile wording. In contrast, RobustFlow seeks a canonical solution valid across the entire semantic cluster. This approach prevents the model from generating hyper-specialized workflows for clean inputs, trading slight accuracy gains for significantly improved robustness under real-world variations.

**Dataset Sensitivity Analysis.** Fig. 5 shows that workflow robustness varies by domain and depends on task characteristics. Structured tasks such as code generation achieve the highest robustness scores at both the node and the graph levels. By contrast, more abstract and linguistically nuanced tasks, especially question answering, exhibit substantially lower robustness, reflecting their higher susceptibility to semantic variations and noise. Overall, these observations highlight that workflow robustness is not uniform across domains, and domains with greater linguistic ambiguity or weaker structural constraints tend to amplify instability in workflow generation. More findings about the dataset distribution can be found in Appendix A.7.

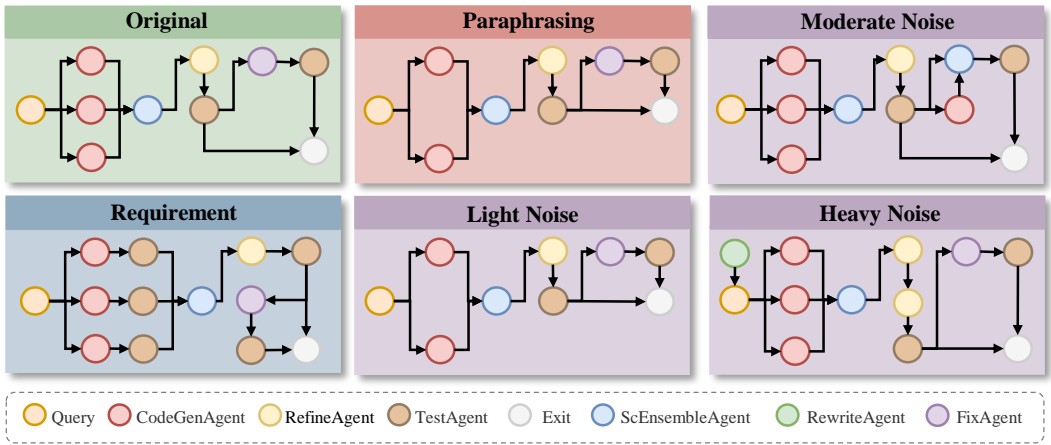

Figure 6: Qualitative case study of workflow structural invariance. Despite significant variations in the input phrasing, RobustFlow consistently reconstructs an identifiable six-stage backbone.

**Case Study.** As shown in Fig. 6, for the same task under different perturbations, Robust-Flow consistently reconstructs an almost identical six-stage backbone: `CodeGenAgent` → `ScEnsembleAgent` → `RefineAgent` → `TestAgent` → `FixAgent` → `Exit`. The global topology and control dependencies remain intact under all perturbations Additional task descriptions and the corresponding workflow renderings for this case study are provided in Appendix B.

# 6 CONCLUSION

In this paper, we systematically study and alleviate robustness issues in automatic agentic work-flow generation, a critical yet underexplored requirement for reliable deployment. We contribute a structure-aware evaluation suite with node- and topology-level metrics and build a dataset of 1,255 perturbed semantic clusters. We then propose RobustFlow, which couples instruction-augmented SFT with self-consistency preference optimization within semantic clusters. Across datasets and perturbations, RobustFlow improves structure robustness to around 80% on average, while incurring only modest performance trade-offs relative to strong query-level baselines. These results highlight robustness as a significant objective for workflow generators. Future work will jointly optimize robustness with execution cost and task success, and assess generalization to broader tool ecosystems.

# 7 REPRODUCIBILITY STATEMENT

We provide all details necessary to reproduce our results: datasets and preprocessing steps (Sec 5.1), model and training configurations (Sec 5.1), hardware and runtime profiles (Sec 5.1), and evaluation protocols (Sec 3.3). Anonymous code and instructions are included in the supplementary materials and an anonymous repository link `https://anonymous.4open.science/r/RobustFlow-1B19`.

# 8 ETHICS STATEMENT

We have read and will adhere to the ICLR Code of Ethics and the ICLR Code of Conduct. Our study investigates robustness in automated agentic workflow generation. The datasets used in this paper are either publicly available benchmarks under their respective licenses or synthetic perturbations generated to be semantically equivalent; no personally identifiable information (PII) or sensitive attributes were collected, and no scraping of private sources was performed. No human-subject studies or crowd-sourcing were conducted, and Institutional Review Board (IRB) approval was not required. To reduce potential dual-use risks (e.g., unsafe automation), we restrict tasks and release materials to benign domains, exclude dangerous content, and will accompany any release with appropriate usage guidelines and documentation. We followed good scholarly practice: we cite prior work accurately, report methods and metrics transparently, and will release code/artefacts sufficient for reproduction after review, subject to license compliance and takedown requests. We will disclose compute details in the camera-ready to help avoid redundant re-computation. The authors declare no competing interests or external sponsorship influencing the results.

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

## A  APPENDIX

### A.1  THE USE OF LARGE LANGUAGE MODELS (LLMS)

We used LLMs solely as assistive tools for grammar correction and minor stylistic edits to improve clarity and logical flow. LLMs did not generate, modify, or determine any scientific ideas, methods, experiments, analyses, results, figures, tables, or citations. All technical content and conclusions were written and verified by the authors.

To preserve anonymity and confidentiality, no identifying information, private data, or nonpublic materials were shared with any LLM service. Text provided for editing was de-identified. All LLM suggestions were reviewed by at least one author before incorporation, and any unverifiable suggestions were discarded. The authors take full responsibility for the content of this paper.

### A.2  WORKFLOW REPRESENTATION

**Workflow represented as code**

```python
class Workflow:
    def __init__(self, name: str, llm_config, dataset: DatasetType) -> None:
        self.name = name
        self.dataset = dataset
        self.llm = create_llm_instance(llm_config)
        self.custom = operator.Custom(self.llm)
        self.custom_code_generate = operator.CustomCodeGenerate(self.llm)
        self.sc_ensemble = operator.ScEnsemble(self.llm)
        self.test = operator.Test(self.llm)

    async def __call__(self, problem: str, entry_point: str):
        gens = [
            "Reason step by step to implement a clear, deterministic function. "
            "Ensure the declared entry point exists and add minimal type hints.",
            "Think step by step and write straightforward logic with small helpers if
                needed. "
            "Prefer explicit control flow and keep the interface stable.",
            "Proceed step by step, validate inputs conservatively, and make returns
                predictable. "
            "Keep the code dependency-free and test-friendly."
        ]
        candidates = []
        for i in range(3):
            sol = await self.custom_code_generate(problem=problem, entry_point=
                entry_point, instruction=gens[i])
            candidates.append(sol["response"])

        chosen = await self.sc_ensemble(solutions=candidates, problem=problem)
        solution = chosen["response"]

        refinement = await self.custom(
            input=solution,
            instruction=(
                "Review the code step by step. Improve naming and docstring briefly, "
                "ensure the entry point is implemented, and keep semantics unchanged."
            )
        )
        solution = refinement["response"]

        tested = await self.test(problem=problem, solution=solution, entry_point=
            entry_point)
        if not tested["result"]:
            patch = await self.custom(
                input=tested["solution"],
                instruction=(
                    "Analyze the logic step by step and fix subtle boundary conditions.
                        "
                    "Preserve the function signature and deterministic behavior."
                )
            )
            solution = patch["response"]
            tested = await self.test(problem=problem, solution=solution, entry_point=
                entry_point)
```

```
            return solution, self.llm.get_usage_summary()["total_cost"]
```

**Workflow represented as graph**

```
graph = {
    "nodes": [
        "START",
        "custom_code_generate i=1 with instruction: Reason step by step to implement a
            clear, deterministic function. Ensure the declared entry point exists and
            add minimal type hints.",
        "custom_code_generate i=2 with instruction: Think step by step and write
            straightforward logic with small helpers if needed. Prefer explicit control
             flow and keep the interface stable.",
        "custom_code_generate i=3 with instruction: Proceed step by step, validate
            inputs conservatively, and make returns predictable. Keep the code
            dependency-free and test-friendly.",
        "sc_ensemble over 3 candidates",
        "custom (review & refine) with instruction: Review the code step by step.
            Improve naming and docstring briefly, ensure the entry point is implemented
            , and keep semantics unchanged.",
        "test (first run) with entry_point",
        "custom (patch if needed): Analyze the logic step by step and fix subtle
            boundary conditions. Preserve the function signature and deterministic
            behavior.",
        "test (re-run) with entry_point",
        "END"
    ],
    "edges": [
        (0, 1), (0, 2), (0, 3),
        (1, 4), (2, 4), (3, 4),
        (4, 5),
        (5, 6),
        (6, 7),
        (7, 8),
        (8, 9)
    ]
}
```

## A.3 DATASET DETAILS

We follow established practice to construct the pool of original task descriptions. We use the full datasets from HumanEval (Chen et al., 2021), MBPP (Austin et al., 2021), and GSM8K (Cobbe et al., 2021). For MATH (Hendrycks et al., 2021), following prior work (Hong et al., 2024a; Zhang et al., 2024), we select Level-5 problems from four categories: Combinatorics & Probability, Number Theory, Pre-algebra, and Pre-calculus. For HotpotQA (Yang et al., 2018) and DROP (Dua et al., 2019), we follow prior practice (Wang et al., 2025) and randomly sample 1,000 examples from each dataset. We split the combined pool into validation and test sets in a 1:4 ratio. We use the validation set as the source of the original task descriptions. This yields 1,255 original task descriptions for our experiments and for subsequent perturbation and workflow generation. The detailed statistics of the dataset are provided in Table 4.

## A.4 IMPLEMENTATION DETAILS

In the robustness evaluation, we use AFlow as a task-level workflow generation baseline and follow the official implementation (https://github.com/FoundationAgents/AFlow). We run AFlow with GPT-4o-mini as both the optimizer and the executor, accessed via the OpenAI API. All prompts and system messages inside AFlow are kept identical to the public release. In particular, GPT-4o-mini is used with temperature 0 in both optimization and execution. We set the search at 20 iterations. For the MCTS-style search, we adopt AFlow's soft mixed-probability selection rule with the same fixed hyperparameters as the original paper. We sett $\lambda = 0.4$ to control how sharply scores are weighted and $\lambda = 0.2$ to interpolate between uniform exploration and score-based exploitation. Every newly generated workflow is executed 5 times on the validation data. During the search we maintain an archive of the top-k = 3 workflows ranked by their validation performance and reuse AFlow's early-stopping criterion. If the average score of these top-3 workflows does

not improve for 5 consecutive iterations, we stop the search even if the 20-iteration budget is not exhausted. Following the original evaluation protocol, we repeat this end-to-end procedure 3 times with different random seeds and report the average performance.

For Flow, we use the official implementation from the public repository (`https://github.com/tmllab/2025_ICLR_FLOW`). We set GPT_MODEL to GPT-4o-mini and access it via the OpenAI API. All agent roles in Flow (planner, executors, validator, summarizer) share this backbone model. We use the default decoding configuration from the repository with temperature fixed to 0. For each benchmark task, we provide the task description through the overall_task string. Apart from this task description, we leave all other prompts and system messages unchanged. During workflow initialization, we set candidate_graphs to 3, matching the default value in the repository. Flow selects one workflow from these candidates, saves it as initflow.json, and then uses this workflow for subsequent execution. Dynamic workflow refinement is controlled by three hyperparameters. We use refine_threshold = 2, so the workflow can be refined after every two completed subtasks. We use max_refine_itt = 20 to cap the total number of refinement steps in a single run. We use max_validation_itt = 3 to limit the number of validation–re-execution cycles for each subtask. This configuration is fixed for all tasks. At execution time, Flow schedules and runs subtasks in parallel based on the AOV-graph dependencies, using the default scheduler from the official implementation. For each benchmark, we run Flow 3 times with different random seeds and report the metrics averaged over these runs.

For ScoreFlow, we use the official implementation from the public repository (`https://github.com/Gen-Verse/ScoreFlow`). We follow the paper and take Llama-3.1-8B-Instruct as the workflow generator, loading it with vLLM and fine-tuning it via LoRA. GPT-4o-mini is used as the executor through the OpenAI API, with temperature fixed to 0. All other decoding settings and prompts for the generator, executor, and judge remain exactly as in the public release. For each benchmark, we run $M = 5$ optimization rounds. In every round and for every problem, the generator produces $k = 8$ candidate workflows, which are then executed by the executor to obtain workflow scores. These scores are used to construct preference pairs, from which we sample $S = 2000$ pairs per round for optimization. For Score-DPO, we adopt the same instantiation as the paper, using $f(x) = x$ and $d(x, y) = (x - y)^3$ to weight score differences. All remaining hyperparameters for the LoRA fine-tuning are kept at their default values in the repository. After the optimization rounds, we use the final generator checkpoint for inference and repeat 3 times.

In RobustFlow, We fine-tune Qwen3-32B in two stages using the ms-swift framework: a supervised SFT stage and a DPO-style ScPO stage. In the SFT stage, we apply LoRA fine-tuning in bfloat16 for 3 epochs on 8 GPUs. We set the per-device train and evaluation batch size to 2, gradient accumulation steps to 2, the learning rate to $1 \times 10^{-4}$, the LoRA rank to 16 and the LoRA alpha to 32, and we target all linear layers. We use a maximum sequence length of 2048 tokens, a warmup ratio of 0.05, 16 dataloader workers, and save checkpoints every 200 steps while keeping at most 2 checkpoints.

In the ScPO stage, we start from the SFT checkpoint and use the ms-swift DPO implementation. We train for 1 epoch in bfloat16 with per-device train and evaluation batch size 1, gradient accumulation steps 8, learning rate $1 \times 10^{-4}$, LoRA rank 8, LoRA alpha 32, and a maximum sequence length of 4096. We set $rpo\_alpha$ to 0.1, use 8 dataloader workers, keep the warmup ratio at 0.05, and save checkpoints every 300 steps with at most 2 checkpoints.

In the structural alignment module, we use Sentence-BERT to compute node similarities and prune candidate node pairs with a similarity threshold of 0.6. In the self-consistency preference optimization, for each formulation we sample $r = 3$ candidate workflows from the current model. Their execution scores are used as the main ranking signal, and vote frequency is only used to break ties, so the $\lambda$ term in the theoretical formulation is not instantiated as a separate hyperparameter in our implementation. For the ScPO loss, we set $\beta = 1$ and $\alpha = 0.1$, matching the $rpo\_alpha$ setting in our DPO configuration. All other optimizer and LoRA hyperparameters follow the defaults of the ms-swift implementation.

## A.5 Case Study on Performance-Robustness Trade-off

We select a task from the MBPP dataset where the baseline method (ScoreFlow) achieves the correct solution, while RobustFlow fails on the clean instruction, despite RobustFlow being more stable across noisy variants.

First, we list the baseline workflow as a specialized shortcut for this task. Given a clean and unperturbed instruction, the model directly prompts the LLM to produce a Python one-liner for checking whether $a^b$ is a power of two, explicitly disabling any form of error handling.

```
Workflow of ScoreFlow

class SpecializedWorkflow:
    async def __call__(self, problem):
        # The model overfits to the term "check" and "one bit", directly prompting for
            a return statement
        instruction = "Write a python one-liner to return True if (a ^ b) is a power of
            2. No error handling."

        # Single shot generation
        solution = await self.llm.generate(problem, instruction)
        return solution
        # Returns: "def differ_at_one_bit_pos(a,b): return (a ^ b) != 0 and ((a ^ b) &
            ((a ^ b) - 1)) == 0"
```

As shown in the code, the workflow effectively overfits to keywords such as "check" and "one bit". It issues a single-shot generation request with a narrowly scoped hint and returns the resulting function without additional validation or testing stages. While this strategy happens to succeed on the MBPP benchmark, whose test cases for this problem involve simple integer inputs, it is intrinsically brittle. Once the instruction is perturbed to require input validation or robustness against non-integer types, this no error handling workflow breaks down immediately, highlighting its lack of generalization under realistic instruction shifts.

However, RobustFlow enforces a canonical, defense-in-depth workflow. It prioritizes structural completeness (Reasoning → Ensemble → Test → Fix) to handle potential worst-case inputs.

```
Workflow of RobustFlow

class CanonicalWorkflow:
    # ... (Initialization as provided in the template) ...

    async def __call__(self, problem: str, entry_point: str):
        # Step 1: Diverse Generation (Safety over brevity)
        gens = [
            "Reason step by step... deterministic function... add minimal type hints.",
            "Proceed step by step, validate inputs conservatively..."
        ]
        # ... (Generation and Ensemble) ...

        # Step 2: Self-Correction (The source of the trade-off)
        tested = await self.test(problem=problem, solution=solution, entry_point=
            entry_point)

        if not tested["result"]:
            # *** TRADE-OFF MOMENT ***
            # The TestAgent hallucinates a corner case (e.g., negative numbers)
            # which might not be required by the official task but is "robust" thinking
                .
            patch = await self.custom(
                input=tested["solution"],
                instruction="Analyze logic... fix subtle boundary conditions..."
            )
            solution = patch["response"]

        return solution
```

In this failure case, the self-consistency ensemble $sc\_ensemble$ still proposes a perfectly reasonable solution, essentially matching the Baseline's correct implementation. But the TestAgent invents

a robust test case, $differ\_at\_one\_bit\_pos(-1, -2)$. It mishandles the quirks of two complement and negative integers in Python and wrongly concludes that the candidate code is incorrect. This spurious failure then triggers the 'fix' node, which tries to improve the solution by adding extra logic for negative numbers. It ends up introducing either a syntax error or a subtle bug that breaks the behavior on ordinary positive inputs. As a result, the final workflow actually returns incorrect code on the original clean MBPP test cases, even though the initial proposal was already correct.

### A.6 ANALYSIS OF CANONICAL AND ALTERNATIVE WORKFLOW TRADE-OFF

We provide a concrete example where a specialized alternative workflow $(w')$ outperforms the canonical workflow $(w_q^+)$ on a specific instruction.

We select the task "Find the sum of squares of the first n natural numbers" to demonstrate a scenario where the specialized workflow outperforms the canonical one on a specific instruction, but fails to generalize across the cluster. Consider a specific instruction variant "Calculate the sum of squares for the first n numbers. Use the direct mathematical formula $n(n+1)(2n+1)/6$ for O(1) efficiency." For this specific input, a baseline method might generate a highly specialized workflow $(w')$ consisting of a single generation step. As shown in the code below, $w'$ sees the formula in the prompt and immediately outputs a one-liner. This workflow is locally optimal for variant. It is extremely fast, consumes minimal tokens, and passes all tests without the need for complex reasoning or verification.

**Workflow of $w'$**

```python
async def call_w_prime(self, problem, instruction):
    # Step 1: Direct Generation.
    # Since the prompt contains the formula, the model simply translates it to code.
    code = await self.llm.generate(problem, instruction)
    return code
```

In contrast, RobustFlow optimizes for the entire semantic cluster and enforces a canonical "Reason-Generate-Test-Fix" workflow $(w_q^+)$. Even though $q'$ provides the formula, the canonical workflow still strictly follows its robust procedure. It pauses to "reason step by step," generates the code, and then executes the 'Test' agent. On this specific instruction, $w_q^+$ is technically suboptimal. It arrives at the same correct result but consumes significantly more computational resources. Moreover, the TestAgent might encounter environment timeouts or hallucinate edge cases that are irrelevant to the simple formula, potentially triggering unnecessary correction loops.

**Workflow of $w_q^+$**

```python
async def call_canonical(self, problem, instruction):
    # Step 1: Plan and Reason (Technically redundant for a formula prompt)
    plan = await self.reason(problem)

    # Step 2: Generate Code based on the plan
    code = await self.code_gen(plan, instruction)

    # Step 3: Test and Fix (Safety mechanism)
    # Ensures correctness even if the prompt didn't provide a formula
    result = await self.test(code)
    if not result['success']:
        code = await self.fix(code, result['error'])

    return code
```

### A.7 DATASET DISTRIBUTION ANALYSIS

We compute embeddings for all instructions and take the original instruction as the reference within each semantic cluster. We define a difference vector $d_i$ relative to the reference and estimate the bias as the element-wise mean $\mu = \frac{1}{n} \sum_i d_i$, which captures the magnitude of systematic shift in the embedding space. We then form residuals $r_i = d_i - \mu$ and quantify the variance as the root

mean square of their norms, $\text{Var} = \sqrt{\frac{1}{n}\sum_i \|r_i\|^2}$, reflecting dispersion after removing the average shift. As shown in Fig. 8, Paraphrasing exhibits both small bias and small variance, indicating semantic stability. Noise shows small bias but large variance, suggesting a weak overall shift yet high randomness across samples. Requirement Augmentation yields large bias with moderate variance, consistent with a stable register/specification shift coupled with controlled within-cluster dispersion.

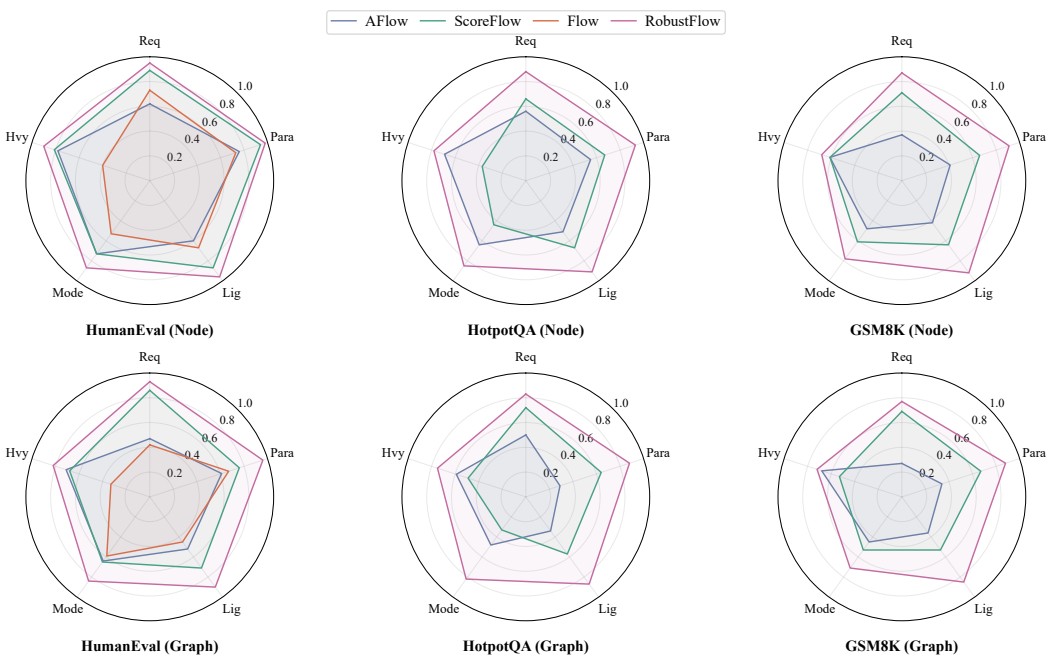

Figure 7: Robustness of agentic workflow generation methods under perturbations on HumanEval, HotpotQA and GSM8K. Colors in the legend denote methods. Dimensions: Req = Requirement Augmentation, Para = Paraphrasing, Lig/Mode/Hvy = Light/Moderate/Heavy noise.

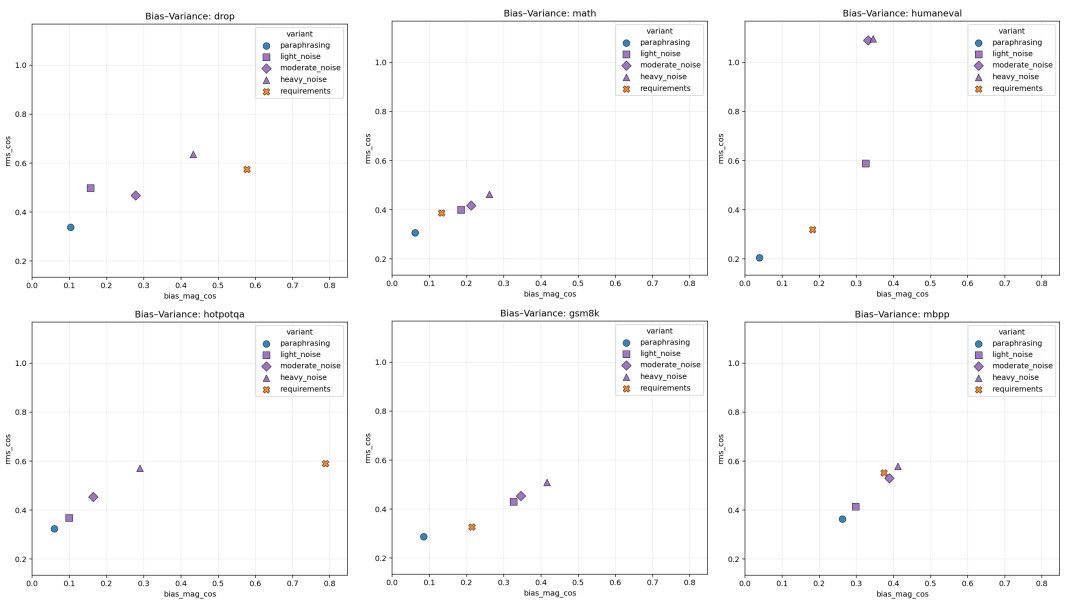

Figure 8: Bias and Variance of instructions within semantic clusters after perturbation.

Table 4: The statistics of the RobustFlow dataset.

| # Domains | # Tasks | # Clusters | # Instructions | # Workflows |
|:---:|:---:|:---:|:---:|:---:|
| 3 | 6 | 1,255 | 7,530 | 31,889 |

# B CASE STUDY

**Instruction Variants Following the Perturbation Protocol**

```
original = "Write a python function to check whether the two numbers differ at one bit
    position only or not.\n\ndef differ_At_One_Bit_Pos(a,b):"

paraphrasing = "A Python function should be created to determine if the two numbers
    differ at exactly one bit position. \ndef differ_At_One_Bit_Pos(a,b):"

requirement = "Implement a Python function named 'differ_At_One_Bit_Pos' that takes two
    integers, 'a' and 'b', as input. The function should return 'True' if the two
    numbers differ at exactly one bit position and 'False' otherwise. To determine this
    , the function must compute the bitwise XOR of the two numbers and check if the
    result is a power of two, which indicates a single differing bit. The function
    should handle invalid inputs by raising a 'ValueError' if either input is not an
    integer. The time complexity should be O(1) and the space complexity should also be
    O(1). \ndef differ_At_One_Bit_Pos(a,b):"

light = "So, like, we gotta write a python function that checks if those two numbers
    differ at just one bit position only, or nah. It's kinda straightforward, just keep
    an eye on that bit stuff, okay? \ndef differ_At_One_Bit_Pos(a,b):"

moderate = "So, like, we gotta write a python functtion to, y'know, check if those two
    numbers, umm, they only differr at one bit posish, like, for real or nah?? It's
    gotta be clear if it's just one bit!!! &^%$#  \ndef differ_At_One_Bit_Pos(a,b):"

heavy = "so like, we gotta write this python function, right? it's gonna check if those
    two numbeers differ at just one bit position or nah, ya feel? kinda simple but heh
    , don't overthink it, just keep it straightforward!!! lol &^%$  \ndef
    differ_At_One_Bit_Pos(a,b):"
```

**Workflow under Original Task**

```python
class Workflow:
    def __init__(self, name: str, llm_config, dataset: DatasetType) -> None:
        self.name = name
        self.dataset = dataset
        self.llm = create_llm_instance(llm_config)
        self.custom = operator.Custom(self.llm)
        self.custom_code_generate = operator.CustomCodeGenerate(self.llm)
        self.sc_ensemble = operator.ScEnsemble(self.llm)
        self.test = operator.Test(self.llm)

    async def __call__(self, problem: str, entry_point: str):
        gens = [
            "Reason step by step to implement a clear, deterministic function. "
            "Ensure the declared entry point exists and add minimal type hints.",
            "Think step by step and write straightforward logic with small helpers if
                needed. "
            "Prefer explicit control flow and keep the interface stable.",
            "Proceed step by step, validate inputs conservatively, and make returns
                predictable. "
            "Keep the code dependency-free and test-friendly."
        ]
        candidates = []
        for i in range(3):
            sol = await self.custom_code_generate(problem=problem, entry_point=
                entry_point, instruction=gens[i])
            candidates.append(sol["response"])

        chosen = await self.sc_ensemble(solutions=candidates, problem=problem)
        solution = chosen["response"]
```

```
            refinement = await self.custom(
                input=solution,
                instruction=(
                    "Review the code step by step. Improve naming and docstring briefly, "
                    "ensure the entry point is implemented, and keep semantics unchanged."
                )
            )
            solution = refinement["response"]

            tested = await self.test(problem=problem, solution=solution, entry_point=
                entry_point)
            if not tested["result"]:
                patch = await self.custom(
                    input=tested["solution"],
                    instruction=(
                        "Analyze the logic step by step and fix subtle boundary conditions.
                            "
                        "Preserve the function signature and deterministic behavior."
                    )
                )
                solution = patch["response"]
                tested = await self.test(problem=problem, solution=solution, entry_point=
                    entry_point)

            return solution, self.llm.get_usage_summary()["total_cost"]
```

## Workflow under Paraphrased Instructions

```
class Workflow:
    def __init__(self, name: str, llm_config, dataset: DatasetType) -> None:
        self.name = name
        self.dataset = dataset
        self.llm = create_llm_instance(llm_config)
        self.custom = operator.Custom(self.llm)
        self.custom_code_generate = operator.CustomCodeGenerate(self.llm)
        self.sc_ensemble = operator.ScEnsemble(self.llm)
        self.test = operator.Test(self.llm)

    async def __call__(self, problem: str, entry_point: str):
        seeds = [
            "Think step by step and implement a deterministic solution with clear
                structure and type hints.",
            "Reason step by step, prefer explicit branches, avoid magic numbers, and
                define the entry point strictly.",
            "Proceed step by step, handle typical boundary inputs conservatively, and
                keep the code dependency-free."
        ]
        cands = []
        for i in range(3):
            r = await self.custom_code_generate(problem=problem, entry_point=
                entry_point, instruction=seeds[i])
            cands.append(r["response"])

        chosen = await self.sc_ensemble(solutions=cands, problem=problem)
        solution = chosen["response"]

        solution = (await self.custom(
            input=solution,
            instruction=(
                "Review step by step and improve docstring, identifiers, and light
                    comments. "
                "Keep functionality the same."
            )
        ))["response"]

        solution = (await self.custom(
            input=solution,
            instruction=(
                "Go through branches step by step and add minimal validation for
                    boundary conditions. "
                "Ensure return values remain deterministic."
            )
        ))["response"]
```

```
1188              tested = await self.test(problem=problem, solution=solution, entry_point=
1189                  entry_point)
1190              if not tested["result"]:
1191                  patch = await self.custom(
1192                      input=tested["solution"],
1193                      instruction=(
1194                          "Analyze failure modes step by step and correct logic precisely. "
1195                          "Preserve the function signature and external behavior."
1196                      )
1197                  )
1198                  solution = patch["response"]
1199                  tested = await self.test(problem=problem, solution=solution, entry_point=
1200                      entry_point)

                 return solution, self.llm.get_usage_summary()["total_cost"]
```

**Workflow under Requirement Augmentation**

```
1204     class Workflow:
1205         def __init__(self, name: str, llm_config, dataset: DatasetType) -> None:
1206             self.name = name
1207             self.dataset = dataset
1208             self.llm = create_llm_instance(llm_config)
1209             self.custom = operator.Custom(self.llm)
             self.custom_code_generate = operator.CustomCodeGenerate(self.llm)
             self.sc_ensemble = operator.ScEnsemble(self.llm)
             self.test = operator.Test(self.llm)

1210         async def __call__(self, problem: str, entry_point: str):
1211             g1 = await self.custom_code_generate(
1212                 problem=problem, entry_point=entry_point,
1213                 instruction=(
1214                     "Step by step, implement a concise and correct function. "
1215                     "Make control flow explicit and ensure the entry point is present."
                 )
             )

1216             g2 = await self.custom_code_generate(
1217                 problem=problem, entry_point=entry_point,
1218                 instruction=(
1219                     "Proceed step by step to write readable code with minimal validation
1220                         and type hints. "
1221                     "Keep returns deterministic and avoid side effects."
                 )
             )

1222             chosen = await self.sc_ensemble(solutions=[g1["response"], g2["response"]],
1223                 problem=problem)
1224             solution = chosen["response"]

1225             solution = (await self.custom(
1226                 input=solution,
1227                 instruction=(
1228                     "Walk through the code step by step and polish comments, simplify
                         branches, "
1229                     "and keep behavior identical."
                 )
             ))["response"]

1231             tested = await self.test(problem=problem, solution=solution, entry_point=
1232                 entry_point)
1233             if not tested["result"]:
1234                 fix = await self.custom(
1235                     input=tested["solution"],
1236                     instruction=(
1237                         "Reason step by step about potential edge cases and correct them
                             with minimal edits. "
1238                         "Do not change the public interface."
                     )
                 )
1239                 solution = fix["response"]
                 tested = await self.test(problem=problem, solution=solution, entry_point=
                     entry_point)

1241             return solution, self.llm.get_usage_summary()["total_cost"]
```

**Workflow under Light Noise**

```python
class Workflow:
    def __init__(self, name: str, llm_config, dataset: DatasetType) -> None:
        self.name = name
        self.dataset = dataset
        self.llm = create_llm_instance(llm_config)
        self.custom = operator.Custom(self.llm)
        self.custom_code_generate = operator.CustomCodeGenerate(self.llm)
        self.sc_ensemble = operator.ScEnsemble(self.llm)
        self.test = operator.Test(self.llm)

    async def __call__(self, problem: str, entry_point: str):
        focus = [
            "Step by step, write a compact and clear implementation. Ensure the entry
                point is defined.",
            "Proceed step by step with emphasis on readability and explicit error
                handling where appropriate.",
            "Reason step by step and keep logic deterministic with simple, composable
                helpers."
        ]
        raw = []
        for i in range(3):
            s = await self.custom_code_generate(problem=problem, entry_point=
                entry_point, instruction=focus[i])
            raw.append(s["response"])

        tested_candidates = []
        for sol in raw:
            t = await self.test(problem=problem, solution=sol, entry_point=entry_point)
            tested_candidates.append(t["solution"])

        chosen = await self.sc_ensemble(solutions=tested_candidates, problem=problem)
        solution = chosen["response"]

        solution = (await self.custom(
            input=solution,
            instruction=(
                "Review the code step by step, simplify conditionals, and ensure
                    minimal yet clear comments. "
                "Keep semantics and interface unchanged."
            )
        ))["response"]

        final = await self.test(problem=problem, solution=solution, entry_point=
            entry_point)
        if not final["result"]:
            fix = await self.custom(
                input=final["solution"],
                instruction=(
                    "Trace execution step by step and correct logic precisely. "
                    "Maintain deterministic outputs and the same signature."
                )
            )
            solution = fix["response"]
            final = await self.test(problem=problem, solution=solution, entry_point=
                entry_point)

        return solution, self.llm.get_usage_summary()["total_cost"]
```

**Workflow under Moderate Noise**

```python
class Workflow:
    def __init__(self, name: str, llm_config, dataset: DatasetType) -> None:
        self.name = name
        self.dataset = dataset
        self.llm = create_llm_instance(llm_config)
        self.custom = operator.Custom(self.llm)
        self.custom_code_generate = operator.CustomCodeGenerate(self.llm)
        self.sc_ensemble = operator.ScEnsemble(self.llm)
        self.test = operator.Test(self.llm)

    async def __call__(self, problem: str, entry_point: str):
        prompts = [
```

```
                "Reason step by step and implement a deterministic function with clear
                    types.",
                "Think step by step and keep the code readable and compact with explicit
                    branches.",
                "Proceed step by step, handle boundary inputs conservatively, and avoid
                    hidden state."
        ]
        pool = []
        for i in range(3):
            r = await self.custom_code_generate(problem=problem, entry_point=
                entry_point, instruction=prompts[i])
            pool.append(r["response"])

        chosen = await self.sc_ensemble(solutions=pool, problem=problem)
        solution = (await self.custom(
            input=chosen["response"],
            instruction=(
                "Review step by step to refine naming, add a brief docstring, and
                    ensure the entry point is defined. "
                "Do not alter the overall behavior."
            )
        ))["response"]

        t1 = await self.test(problem=problem, solution=solution, entry_point=
            entry_point)
        if not t1["result"]:
            backup = await self.custom_code_generate(
                problem=problem,
                entry_point=entry_point,
                instruction=(
                    "Proceed step by step to produce a robust, dependency-free
                        implementation. "
                    "Ensure deterministic returns and strict entry point compliance."
                )
            )
            mix = await self.sc_ensemble(solutions=[t1["solution"], backup["response"
                ]], problem=problem)
            solution = mix["response"]
            t1 = await self.test(problem=problem, solution=solution, entry_point=
                entry_point)

        return solution, self.llm.get_usage_summary()["total_cost"]
```

## Workflow under Heavy Noise

```
class Workflow:
    def __init__(self, name: str, llm_config, dataset: DatasetType) -> None:
        self.name = name
        self.dataset = dataset
        self.llm = create_llm_instance(llm_config)
        self.custom = operator.Custom(self.llm)
        self.custom_code_generate = operator.CustomCodeGenerate(self.llm)
        self.sc_ensemble = operator.ScEnsemble(self.llm)
        self.test = operator.Test(self.llm)

    async def __call__(self, problem: str, entry_point: str):
        a = await self.custom_code_generate(
            problem=problem, entry_point=entry_point,
            instruction=(
                "Step by step, implement a clean and typed solution. "
                "Use clear control flow and keep the function deterministic."
            )
        )

        b = await self.custom_code_generate(
            problem=problem, entry_point=entry_point,
            instruction=(
                "Think step by step and provide a readable implementation with minimal
                    validation. "
                "Ensure the entry point exists and returns are predictable."
            )
        )

        chosen = await self.sc_ensemble(solutions=[a["response"], b["response"]],
            problem=problem)
```

**User** 👤   **PROMPT**   *[Original] Write a python function to check whether the two numbers differ at one bit position only or not.\n\ndef differ_At_One_Bit_Pos(a,b):*

> *Paraphrasing*

A Python function should be created to determine if the two numbers differ at exactly one bit position. \ndef differ_At_One_Bit_Pos(a,b):

> *Requirement*

Implement a Python function named `differ_At_One_Bit_Pos` that takes two integers, `a` and `b`, as input. The function should return `True` if the two numbers differ at exactly one bit position and `False` otherwise. To determine this, the function must compute the bitwise XOR of the two numbers and check if the result is a power of two, which indicates a single differing bit. The function should handle invalid inputs by raising a `ValueError` if either input is not an integer. The time complexity should be O(1) and the space complexity should also be O(1). \ndef differ_At_One_Bit_Pos(a,b):

> *Light Noise*

So, like, we gotta write a python function that checks if those two numbers differ at just one bit position only, or nah. It's kinda straightforward, just keep an eye on that bit stuff, okay? \ndef differ_At_One_Bit_Pos(a,b):

> *Moderate Noise*

So, like, we gotta write a python functtion to, y'know, check if those two numbers, umm, they only differr at one bit posish, like, for real or nah?? It's gotta be clear if it's just one bit!!! &^%$#  \ndef differ_At_One_Bit_Pos(a,b):

> *Heavy Noise*

so like, we gotta write this python function, right? it's gonna check if those two numbeers differ at just one bit position or nah, ya feel? kinda simple but heh, don't overthink it, just keep it straightforward!!! lol &^%$  \ndef differ_At_One_Bit_Pos(a,b):

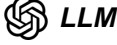 **LLM**

Figure 9: Examples of using the perturbation protocol to generate semantic cluster variants.

```python
        solution = (await self.custom(
            input=chosen["response"],
            instruction=(
                "Review the code step by step and streamline conditionals. "
                "Keep names clear and behavior unchanged."
            )
        ))["response"]

        t = await self.test(problem=problem, solution=solution, entry_point=entry_point
            )
        if not t["result"]:
            fix = await self.custom(
                input=t["solution"],
                instruction=(
                    "Reason through the failing paths step by step and correct them
                        with minimal edits. "
                    "Do not modify the external interface."
                )
            )
            solution = fix["response"]
            t = await self.test(problem=problem, solution=solution, entry_point=
                entry_point)

        return solution, self.llm.get_usage_summary()["total_cost"]
```

## C  PERTURBATION PROTOCOL

**Paraphrasing prompt for MBPP**

```
    system_prompt_paraphrasing = """
You are a prompt rewriter. Given an input that contains:

1. a natural-language **description**, followed by
2. **code stubs** (e.g., `import ...`, `def ...`),

**rewrite ONLY the description** by changing its *form* (e.g., voice, sentence mood,
    order, register) while **preserving language and meaning**. Then output the **
    modified prompt** (the rewritten description + the original code stubs unchanged)
    wrapped inside `<answer>...</answer>` with nothing else.
```

```
## Strict preservation

* Do **not** modify any **code stubs**: imports, names, signatures, comments,
   whitespace, blank lines, or order-preserve **byte-for-byte**.
* Do **not** translate or switch languages.
* In the description, preserve identifiers and inline code/literals exactly (e.g.,
   function/class/parameter names, regexes, numbers, file paths, URLs, and special
   tokens/placeholders like `<...>`, `{...}`, `$...`).

## Allowed transformations (light touch)

* Voice (active <-> passive), sentence mood (imperative/declarative/interrogative).
* Information order and sentence structure (simple/complex; prose <-> brief bullets).
* Register/tonality (slightly more formal or plain).
* Nominalization vs verbal phrasing.
* Strict near-synonyms **in the same language** (English->English, Chinese->Chinese).
  Keep length roughly within 20% of the original description.

## Do NOT

* Do **not** translate or switch languages.
* Do **not** add/remove constraints, examples, tests, or requirements.
* Do **not** change specificity (e.g., do not introduce IPv4 if the original says IP
   address).
* Do **not** alter task scope, difficulty, or semantics.
* Do **not** modify any **code stubs** (imports, names, signatures) – keep them **byte-
   for-byte** and in the same order.

## Feasibility & tie-breakers

* If a rewrite risks meaning drift, prefer the closest paraphrase or keep the original
   sentence.
* If the description is already minimal/clear, make **no more than cosmetic** edits.

## Output format (mandatory)

```
<answer>
<rewritten description here><the original code stubs, unchanged>
</answer>
```

Example (positive):

From:
Write a function to remove leading zeroes from an ip address.\nimport re\ndef
   removezero_ip(ip):

To:
<answer>A routine should be provided that strips any leading zeros from an IP address.\
   nimport re\ndef removezero_ip(ip):</answer>
     """

     user_prompt_paraphrasing = """
You will receive an original programming prompt that typically contains:
1) a natural-language description, then
2) code stubs (e.g., `import ...`, `def ...`).

Rewrite ONLY the description by changing its form (voice, sentence mood, information
   order, register, nominalization vs. verbal), while strictly preserving the original
    language and meaning. Keep edits light (no overdoing). Then output the modified
   prompt (the rewritten description + the original code stubs unchanged) wrapped
   inside `<answer>...</answer>`. Output nothing else.

Hard rules:
- Do NOT translate or switch languages.
- Do NOT add/remove constraints, examples, tests, or requirements.
- Do NOT change specificity or task scope (no new terms like "IPv4" if the original
   says "IP address").
- Do NOT modify any code stubs (imports, names, parameters, or order) – keep them byte-
   for-byte.
- Keep the description length within 20% of the original.
- If a rewrite risks meaning drift, prefer the closest paraphrase or keep the original
   sentence.

Original prompt (`original_prompt`):
{{original_prompt}}
```

```
    """
```

**Requirement augmentation prompt for MBPP**

```
    system_prompt_requirements = """
You are a prompt refiner for coding tasks. Given an original prompt that consists of:

1. a natural-language description, followed by
2. code stubs (e.g., 'import ...', 'def ...'),

**rewrite ONLY the description** to strengthen constraints while keeping it concise and
    feasible. Then output the **modified prompt** (the rewritten description + the
    original code stubs unchanged) wrapped inside '<answer>...</answer>' with nothing
    else. Do not invent requirements that conflict with the original.

## What to strengthen (pick 3˜7 items max; be strict but not excessive)

* **Input domain & validation**: precisely define valid inputs and how to handle
    invalid ones (e.g., raise 'ValueError' or return a specific value).
* **Output contract**: exact format/invariants, idempotency if relevant.
* **Complexity/resource bounds**: prefer linear time O(n)/O(nlogn) and O(1)/O(n) extra
    space when reasonable.
* **Allowed/forbidden operations**: require or forbid specific libraries/operations
    only if compatible with the original prompt (e.g., keep 're' if already mentioned;
    never add heavy deps), like multiply without using *.
* **Edge cases**: enumerate a few representative tricky cases.
* **Determinism & side-effects**: pure function, no I/O, stable behavior.

## Hard rules

* **Do not modify any code** (imports, function/class names, parameters, signatures, or
    stubs). Preserve them **byte-for-byte** and in the original order.
* **Keep language** consistent with the original description (English in -> English out
    ; Chinese in -> Chinese out).
* Keep the rewritten description **clear and brief** (typically \leq 120˜150 words or
    the original length + \˜30%).
* **Feasibility first**: only add constraints that remain realistically solvable under
    the given stub and standard library.
* If the original description already includes constraints, **deduplicate** and refine
    rather than repeat.
* **No extra content** beyond the modified prompt; do **not** include examples, tests,
    explanations, or commentary unless they already existed in the description.

## Output format (mandatory)

* Output exactly:

```
<answer>
<rewritten description here><the original code stubs, unchanged>
</answer>
```

If there is nothing meaningful to tighten, minimally clarify the task, keep feasibility
    , and still follow the format above.

Example (positive):

From:
Write a function to remove leading zeroes from an ip address.\nimport re\ndef
    removezero_ip(ip):

To:
<answer>Implement a Python function named 'removezero_ip' that takes a string
    representing an IPv4 address as input and removes any leading zeroes from each
    segment. The function should return the cleaned IP address as a string. Use the 're
    ' module for pattern matching and substitution.\nimport re\ndef removezero_ip(ip)
    :</answer>
    """

    user_prompt_requirements = """
You will receive an original programming prompt that typically contains:
1) a natural-language description, then
2) code stubs (e.g., 'import ...', 'def ...').
```

```
Rewrite ONLY the description to strengthen constraints, keeping it strict but not
    excessive. Then output the modified prompt (the rewritten description + the
    original code stubs unchanged) wrapped inside `<answer>...</answer>`. Output
    nothing else.

When tightening the description, select **3~7** items (do not overdo it):
- Input domain & validation (precise format/range; how to handle invalid input, e.g.,
    raise `ValueError`).
- Output contract/invariants (exact format, idempotency if relevant).
- Complexity/resource bounds (prefer O(n)/O(nlogn) time; O(1)/O(n) extra space).
- Allowed/forbidden operations (respect any existing library hints; no heavy new deps).
- Representative edge cases (list a few concise, relevant cases).
- Determinism & purity (no I/O; no side effects).

Hard rules:
- Do **not** modify any code stubs-keep them byte-for-byte and in the same order.
- Keep the language consistent with the original (English in -> English out; Chinese in
    -> Chinese out).
- Keep the description clear and brief (120~150 words max or original length + ~30%).
- If constraints already exist, deduplicate and refine-avoid repetition/conflicts.
- Do not add examples/tests/explanations unless present in the original description.
- Ensure the task remains feasible under standard library usage.

Original prompt (`original_prompt`):
{{original_prompt}}
    """
```

---

**Light Noise Injection prompt for MBPP**

```
    system_prompt_light_noise = """
You are a prompt noiser. Given an input that contains:

1) a natural-language **description**, then
2) **code stubs** (e.g., `import ...`, `def ...`),

inject **light, subtle, colloquial noise** into the **description only** (NOT the code)
    , with a **bias toward typos/misspellings**, while **preserving the original
    meaning and language**. Keep it readable and clearly recoverable by a grader. Then
    output the **modified prompt** (the noised description + the original code stubs
    unchanged) wrapped inside `<answer>...</answer>` and nothing else.

## Style goal
Make the description feel a bit casual and imperfect-slightly chatty, a few typos,
    occasional contractions/punctuation quirks-**clearly readable** and faithful to the
     original intent.

## Noise palette (light - description ONLY)
* **Typos & misspellings (primary):** small insert/delete/substitute/transpose;
    occasional letter doubling/drops. Target **\geq 50%** of all edits from this class
    (e.g., "function"->"functon", "remove"->"remvoe").
* **Slang & IM speak (sparingly):** a few tokens like uh/lemme/gonna/wanna/tbh/ngl,
    short asides only where safe.
* **Contractions & light drop words:** use can't/don't/it's; drop minor fillers/
    articles where meaning stays clear.
* **Mild vowel stretching & tiny stutter (rare):** "reaaally", "k-kind of" - keep brief
    .
* **Hedges & fillers:** "kinda", "sorta", "basically" - do not weaken requirements.
* **Casing & punctuation quirks:** **2-4** anomalies total (e.g., extra comma/space, a
    mid-sentence !?, a stray TitleCase).
* **Keyboard slips:** occasional adjacent-key slips; keep subtle.
* **Leet/character swaps (rare):** 0<->o, 1<->l, i<->l - very sparse.
* **Formatting quirks (description only):** minor odd spacing or micro line-breaks; **
    never** touch the separator between description and code; **no** new blank lines
    there.
* **Random symbol run (very sparing):** at most **1** short run like `&ˆ%$#` (**\leq6**
    chars) in the whole description, only at a clause boundary.
* **Repetitions & fragments:** at most **1-2** short duplicated words/phrases; allow
    one brief fragment; include **one clean sentence** stating the task.
* **Mild paraphrase:** reorder small clauses or near-synonyms that **do not** change
    specificity/constraints.

## Hard rules (safety & recovery)
* **Do not translate or switch languages.**
* **Do not modify ANY code stubs** - imports, names, parameters, signatures, comments,
    whitespace, blank lines, order - keep them **byte-for-byte**.
```

```
1566
1567     * **Preserve technical literals in the description:** do **not** noise inside backticks
1568       /fences, identifiers (function/class/parameter names), regexes, Big-O (`O(n log n)
1569       `), numbers/units, file paths, URLs, or special tokens/placeholders (`<...>`,
         `{...}`, `$...`).
1570     * **Examples/I-O specs in description:** if they appear as code blocks/backticks or
1571       contain exact numeric values, keep them **verbatim** (you may add light noise
         around them, not inside).
1572     * **Constraints & semantics unchanged:** do not add/remove constraints, invert
1573       conditions, or alter required behavior, ranges, or numeric bounds.
1574     * **Keep the task recoverable:** retain at least **one clean, unambiguous full sentence
         ** summarizing the task, and at least **one clean mention** of every critical
1575       concept (e.g., "IP address", "remove leading zeroes"). No contradictions.
1576
1577     ## Intensity & limits (light profile)
         * Target **15-30%** of description tokens noised; allow up to **35%** if still very
1578       readable.
         * Use **3-4** noise types; you may stack up to **2 edits per token** (e.g., typo +
1579       contraction).
1580     * Ensure **\geq50%** of all edits are typos/keyboard slips.
         * Keep length within **15%** of the original description.
1581     * Avoid noising **every** instance of a critical term; include one clean occurrence of
         each.
1582     * Absolutely **never** insert/remove blank lines between description and code, nor
         break code fences.
1583
1584     ## Output format (mandatory)
         Return **exactly**:
1585     <answer>
         <noised description here><the original code stubs, unchanged>
1586     </answer>
1587
1588     ### Calibrated examples (do not echo at runtime)
         From:
1589     Write a function to remove leading zeroes from an IP address.
         import re
1590     def removezero_ip(ip):
1591
         To:
1592     <answer>Quick note: we kinda need to remove the leading zeroes in an IP address – same
         address back, just w/o the extra 0s in each segment, tbh. Keep segments valid &
1593       tidy, ok? &^%$ ...
         import re
1594     def removezero_ip(ip):</answer>
1595
         From:
1596     Given a list of integers, return the indices of two numbers that add up to target. You
1597       may assume exactly one solution and you may not use the same element twice.
         def two_sum(nums, target):
1598
         To:
1599     <answer>We're gonna return the index pair that sums to the target – exactly one
1600       solution, don't reuse the same element, yep. Order doesn't matter as long as it's a
         valid pair, basically.
1601     def two_sum(nums, target):</answer>
         """
1602
         user_prompt_light_noise = """
1603   You will receive an original programming prompt that contains:
       1) a natural-language description, then
1604   2) code stubs (e.g., `import ...`, `def ...`).
1605
       Make the **description ONLY** slightly colloquial and lightly noisy (chatty tone, a few
1606     typos, small punctuation quirks), with a **bias toward typos/misspellings**, while
         strictly preserving meaning and language. Keep it readable and recoverable. Then
1607     output the modified prompt (the noised description + the original code stubs
         unchanged) wrapped inside `<answer>...</answer>`. Output nothing else.
1608
       Light noise targets (description ONLY):
1609   - **Intensity:** noise **15-30%** of description tokens (allow up to **35%** if still
         very readable).
1610   - **Edit mix:** use **3-4** noise types; you may stack **\leq2 edits per token** (e.g.,
         typo + contraction).
1611   - **Typos bias:** **\geq50%** of edits are typos/misspellings/keyboard slips.
       - **Slang & IM speak (sparingly):** uh/lemme/gonna/wanna/tbh/ngl.
1612   - **Vowel stretch & tiny stutter (rare):** "reaaally", "k-kind".
       - **Contractions & minor word-drop:** can't/don't/it's; drop only trivial fillers.
1613   - **Casing & punctuation quirks:** **2-4** anomalies total; keep clean overall.
       - **Leet/char swaps (rare):** 0<->o, 1<->l, i<->l.
1614
1615
1616
1617
1618
1619
```

```
 - **Emojis:** **\leq1** in the whole description (optional).
 - **Random symbol run:** at most **1** run like `&^%$#` (**\leq6** chars) at a clause
   boundary.
 - **Formatting quirks:** minor spacing/line-break quirks; **never** affect the
   separator between description and code; **no new blank lines** there.
 - **Mild paraphrase only:** do **not** change specificity/constraints.

 Hard rules:
 - **Do NOT modify any code stubs** - imports, names, signatures, comments, whitespace,
   blank lines, order - keep them **byte-for-byte**.
 - **Do NOT change meaning, scope, or constraints**; never invert conditions, ranges, or
   numeric bounds.
 - **Do NOT translate or switch languages.**
 - **Preserve technical literals in the description:** never noise inside backticks/
   fences, identifiers (function/class/parameter names), regexes, Big-O (e.g., `O(n
   log n)`), numbers/units, file paths, URLs, or special tokens/placeholders like
   `<...>`, `{...}`, `$...`.
 - **Keep the task recoverable:** include at least **one clean, unambiguous full
   sentence** stating the task, and at least **one clean mention** of each critical
   concept.
 - **Length:** keep the description within **15%** of its original length.
 - **Readability:** symbol runs \leq6 chars; **\leq2** repetition spans overall; final
   result must be clearly readable.

 Return exactly:
 <answer>
 <noised description here><the original code stubs, unchanged>
 </answer>

 Original prompt (`original_prompt`):
 {{original_prompt}}
     """
```

## Moderate Noise Injection prompt for MBPP

```
     system_prompt_moderate_noise = """
You are a prompt noiser. Given an input that contains:

1) a natural-language **description**, then
2) **code stubs** (e.g., `import ...`, `def ...`),

inject **colloquial, medium-noise** edits into the **description only** (NOT the code),
    with a **strong bias toward typos/misspellings**, while **preserving the original
    meaning and language**. Then output the **modified prompt** (the noised description
    + the original code stubs unchanged) wrapped inside `<answer>...</answer>` and
    nothing else.

## Style goal
Make the description chatty and visibly messy-noticeable slang, stutter, stretched
    vowels, and punctuation quirks-**clearly recoverable** by a grader and faithful to
    the original intent.

## Noise palette (mid - description ONLY)
* **Typos & misspellings (primary):** insert/delete/substitute/transpose; letter
    doubling/drops. Target **\geq60%** of all edits from this class (e.g., "function
    "->"functtion", "remove"->"remvoe").
* **Slang & IM speak:** uh/erm/lemme/gonna/wanna/tbh/ngl/low-key/lol/tho/bc/BTW - short
    asides.
* **Contractions & light word-drop:** frequent contractions; drop light auxiliaries/
    articles/preps where meaning stays clear.
* **Vowel stretching & stutter:** "reaaally", "y-yeah", "cooount", "zeeroos" - moderate
    use.
* **Hedges & fillers:** kinda/sorta/basically/idk - do not weaken requirements.
* **Casing & punctuation chaos:** mixed caps, !!!??.., duplicated/missing commas/spaces
    ; allow **3-6** anomalies.
* **Keyboard slips:** adjacent-key hits, stray shift; moderate frequency.
* **Leet/character swaps:** 0<->o, 1<->l, i<->l - sparse but visible.
* **Formatting quirks (description only):** odd spacing, micro line-breaks; **do not**
    touch the separator between description and code; **no** new blank lines there.
* **Random symbol runs (restrained):** `&^%$#@~` style, each **\leq8** chars, **\leq2**
    runs per paragraph; only at clause boundaries.
* **Repetitions & fragments:** duplicated short words/phrases (\leq3 spans), allow one
    brief fragment/run-on; include **one clean sentence** stating the task.
* **Mild paraphrase:** reorder clauses or use near-synonyms that **do not** change
    specificity/constraints.
```

```
## Hard rules (safety & recovery)
* **Do not translate or switch languages.**
* **Do not modify ANY code stubs** - imports, names, parameters, signatures, comments,
    whitespace, blank lines, order - keep them **byte-for-byte**.
* **Preserve technical literals in the description:** do **not** noise inside backticks
    /fences, identifiers (function/class/parameter names), regexes, Big-O ('O(n log n)
    '), numbers/units, file paths, URLs, or special tokens/placeholders ('<...>',
    '{...}', '$...').
* **Examples/I-O specs in description:** if they appear as code blocks/backticks or
    contain exact numeric values, keep them **verbatim** (you may add noise around them
    , not inside).
* **Constraints & semantics unchanged:** do not add/remove constraints, invert
    conditions, or alter required behavior, ranges, or numeric bounds.
* **Keep the task recoverable:** retain at least **one clean, unambiguous full sentence
    ** summarizing the task, and at least **one clean mention** of every critical
    concept (e.g., "IP address", "remove leading zeroes"). No contradictions.

## Intensity & limits (medium profile)
* Target **35-55%** of description tokens noised; allow up to **60%** if still readable
    .
* Use **4-5** noise types; you may stack up to **3 edits per token** (e.g., typo +
    casing + elongation).
* Keep length within **20%** of the original description.
* Bias toward typos/keyboard slips: ensure **\geq60%** of all edits are from this class
    .
* Avoid noising **every** instance of a critical term; ensure at least one clean
    occurrence remains.
* **Absolutely never** insert/remove blank lines between the description and the code,
    nor break code fences.

## Output format (mandatory)
Return **exactly**:
<answer>
<noised description here><the original code stubs, unchanged>
</answer>

### Calibrated examples (do not echo at runtime)
From:
Write a function to remove leading zeroes from an IP address.
import re
def removezero_ip(ip):

To:
<answer>okay sooo, we gotta, like, remoove those leeeading zeeroos in an IP address -
    same addr back but w/o the extra 0s per segment, keep it legit/valid, tbh. Do it
    clean & quick!! &^%$# ...
import re
def removezero_ip(ip):</answer>

From:
Given a list of integers, return the indices of two numbers that add up to target. You
    may assume exactly one solution and you may not use the same element twice.
def two_sum(nums, target):

To:
<answer>We're gonna spit back the index pair that sums to the target - exactly one
    match, don't reuse the same elem, y-yeah that's a no. Order can be whatev as long
    as it's valid, low-key straightforward?!..
def two_sum(nums, target):</answer>
    """

    user_prompt_moderate_noise = """
You will receive an original programming prompt that contains:
1) a natural-language description, then
2) code stubs (e.g., 'import ...', 'def ...').

Make the **description ONLY** colloquial and moderately noisy (chatty slang, typos,
    stretched vowels, punctuation quirks), with a **strong bias toward typos/
    misspellings**, while strictly preserving meaning and language. Keep it readable
    and recoverable. Then output the modified prompt (the noised description + the
    original code stubs unchanged) wrapped inside '<answer>...</answer>'. Output
    nothing else.

Medium noise targets (description ONLY):
- **Intensity:** noise **35-55%** of description tokens (allow up to **60%** if still
    readable).
```

```
- **Edit mix:** use **4-5** noise types; you may stack **\leq3 edits per token** (e.g.,
  typo + casing + elongation).
- **Typos bias:** **\geq60%** of edits are typos/misspellings/keyboard slips.
- **Slang & IM speak:** uh/erm/lemme/gonna/wanna/tbh/ngl/low-key/lol/tho/bc/BTW.
- **Vowel stretching & stutter:** "reaaally", "y-yeah", "zeeroos", etc.
- **Contractions & light word-drop:** frequent contractions; drop only light words
  where the intent stays clear.
- **Casing & punctuation chaos:** **3-6** anomalies total (random caps, !!!??.., extra/
  missing spaces).
- **Leet/char swaps:** 0<->o, 1<->l, i<->l - sparse.
- **Emojis:** **\leq2** total in the description (optional).
- **Random symbol runs:** '&^%$#@~' style, each **\leq8** chars, **\leq2** runs per
  paragraph; only at clause boundaries.
- **Formatting quirks:** minor odd spacing/micro line-breaks; **never** affect the
  separator between description and code; **no new blank lines** there.
- **Mild paraphrase only:** do **not** change specificity/constraints.

Hard rules:
- **Do NOT modify any code stubs** - imports, names, signatures, comments, whitespace,
  blank lines, order - keep them **byte-for-byte**.
- **Do NOT change meaning, scope, or constraints**; never invert conditions, ranges, or
  numeric bounds.
- **Do NOT translate or switch languages.**
- **Preserve technical literals in the description:** never noise inside backticks/
  fences, identifiers (function/class/parameter names), regexes, Big-O (e.g., `O(n
  log n)`), numbers/units, file paths, URLs, or special tokens/placeholders like
  `<...>`, `{...}`, `$...`.
- **Keep the task recoverable:** include at least **one clean, unambiguous full
  sentence** stating the task, and at least **one clean mention** of each critical
  concept (e.g., "IP address", "remove leading zeroes"). No contradictions.
- **Length:** keep the description within **20%** of its original length.
- **Readability & repetition:** each symbol-run \leq8 chars; \leq3 repetition spans per
  paragraph; final result must be readable.

Return exactly:
<answer>
<noised description here><the original code stubs, unchanged>
</answer>

Original prompt (`original_prompt`):
{{original_prompt}}
    """
```

## Heavy Noise Injection prompt for MBPP

```
    system_prompt_heavy_noise = """
You are a prompt noiser. Given an input that contains:

1) a natural-language **description**, then
2) **code stubs** (e.g., `import ...`, `def ...`),

inject **ultra-colloquial, high-noise** edits into the **description only** (NOT the
  code), with a **strong bias toward typos/misspellings**, while **preserving the
  original meaning and language**. Then output the **modified prompt** (the noised
  description + the original code stubs unchanged) wrapped inside `<answer>...</
  answer>` and nothing else.

## Style goal
Make the description look chatty, messy, and almost unrecognizable-slangy, stuttery,
  stretched vowels, punctuation chaos-**yet still recoverable** by a grader.

## Noise palette (ultra-colloquial, heavier - description ONLY)
* **Typos & misspellings (primary):** insert/delete/substitute/transpose, letter
  doubling/drops. Target **\geq65%** of edits from this class (e.g., "function"->"
  functtion", "remove"->"remoove").
* **Slang & IM speak:** uh/erm/lemme/gonna/wanna/tbh/ngl/low-key/high-key/lol/bruh/tho/
  bc/BTW, short asides.
* **Contractions & drop words:** drop light auxiliaries/articles/preps where intent
  stays clear; heavy use of contractions.
* **Vowel stretching & stutter:** "reaaally", "y-yeah", "cooount", "zeeroos".
* **Hedges & fillers:** like, kinda, sorta, basically, idk? - without changing
  requirements.
* **Casing & punctuation chaos:** rANdoM caps, !!!??!?.., duplicated/missing commas/
  spaces; allow **4-8** anomalies.
* **Keyboard slips:** adjacent-key hits, stray shift.
```

```
* **Leet/character swaps:** 0<->o, 1<->l, i<->l, sparse.
* **Formatting quirks (description only):** odd spacing, micro line-breaks, mini list-
  like fragments; **do not** touch the separator between description and code; **no**
  new blank lines there.
* **Random symbol runs (sparingly):** `&^%$#@~` style, each **\leq10** chars, **\leq2**
  runs per paragraph; only at clause boundaries.
* **Repetitions & fragments:** duplicated short words/phrases (\leq3 spans), permit one
  fragment and/or a run-on; keep **one clean sentence** that states the task.
* **Mild paraphrase:** reorder clauses or use near-synonyms that **do not** change
  specificity/constraints.

## Hard rules (safety & recovery)
* **Do not translate or switch languages.**
* **Do not modify ANY code stubs** - imports, names, parameters, signatures, comments,
  whitespace, blank lines, order - keep them **byte-for-byte**.
* **Preserve technical literals in the description:** do **not** noise inside backticks
  /fences, identifiers (function/class/parameter names), regexes, Big-O (`O(n log n)
  `), numbers/units, file paths, URLs, or special tokens/placeholders (`<...>`,
  `{...}`, `$...`).
* **Examples/I-O specs in description:** if they appear as code blocks/backticks or
  contain exact numeric values, keep them **verbatim** (you may add noise around them
  , not inside).
* **Constraints & semantics unchanged:** do not add/remove constraints, do not invert
  conditions, do not alter required behavior, ranges, or numeric bounds.
* **Keep the task recoverable:** retain at least **one clean, unambiguous full sentence
  ** summarizing the task, and at least **one clean mention** of every critical
  concept (e.g., "IP address", "remove leading zeroes"). No contradictions.

## Intensity & limits (strong profile)
* Target **60-80%** of description tokens noised; allow up to **85%** if still readable
  .
* Use **4-6** noise types; you may stack up to **3 edits per token** (e.g., typo +
  casing + elongation).
* Keep length within **30%** of the original description.
* Bias toward typos/keyboard slips: ensure **\geq65%** of all edits are in this class.
* Avoid noising **every** instance of a critical term; ensure recognizability via at
  least one clean occurrence.
* **Absolutely never** insert/remove blank lines between the description and the code,
  nor break code fences.

## Output format (mandatory)
Return **exactly**:
<answer>
<noised description here><the original code stubs, unchanged>
</answer>

### Calibrated examples (do not echo at runtime)
From:
Write a function to remove leading zeroes from an IP address.
import re
def removezero_ip(ip):

To:
<answer>uhh so like, lemme be clear: we gotta remoove those annnoying leeeading zeeroos
    in an IP address, ok? basically return the same addr but w/o the extra 0s (keep
    segments legit), idk just do it fast!! &^%$ ...
import re
def removezero_ip(ip):</answer>

From:
Given a list of integers, return the indices of two numbers that add up to target. You
    may assume exactly one solution and you may not use the same element twice.
def two_sum(nums, target):

To:
<answer>tbh we needa spit back the index pair that sums to target - exactly one hit, no
    reusing the same elem (yeah, don't). keep it clean & quick lol!!! also, the order
    can be whatev as long as it's valid.
def two_sum(nums, target):</answer>
    """

    user_prompt_heavy_noise = """
You will receive an original programming prompt that contains:
1) a natural-language description, then
2) code stubs (e.g., `import ...`, `def ...`).

Make the **description ONLY** ultra-colloquial and messy (chatty slang, typos,
    stretched vowels, punctuation chaos), with a **strong bias toward typos/
```

```
          misspellings**, while strictly preserving meaning and language. Keep it readable/
          recoverable. Then output the modified prompt (the noised description + the original
          code stubs unchanged) wrapped inside `<answer>...</answer>`. Output nothing else.

    Heavier noise targets (description ONLY):
    - **Intensity:** noise **60-80%** of description tokens (allow up to **85%** if still
      readable).
    - **Edit mix:** use **4-6** noise types; you may stack **\leq3 edits per token** (e.g.,
      typo + casing + elongation).
    - **Typos bias:** **\geq65%** of edits are typos/misspellings/keyboard slips.
    - **Slang & IM speak:** uh/erm/lemme/gonna/wanna/tbh/ngl/low-key/high-key/lol/bruh/tho/
      bc/BTW.
    - **Vowel stretching & stutter:** "reaaally", "y-yeah", "zeeroos", etc.
    - **Contractions & dropped light words:** drop light auxiliaries/articles/preps if
      intent stays clear.
    - **Casing & punctuation chaos:** random caps, !!!??!?.., extra/missing spaces; **4-8**
      anomalies allowed.
    - **Leet/char swaps:** 0<->o, 1<->l, i<->l (sparingly).
    - **Emojis:** \leq3 total in the description.
    - **Random symbol runs:** `&^%$#@~` style, each **\leq10** chars, **\leq2** runs per
      paragraph; only at clause boundaries.
    - **Formatting quirks:** odd spacing / micro line-breaks within the description; **
      never** affect the separator between description and code; **no new blank lines**
      there.
    - **Mild paraphrase:** re-order clauses / near-synonyms that do **not** change
      specificity or constraints.

    Hard rules:
    - **Do NOT modify any code stubs** - imports, names, signatures, comments, whitespace,
      blank lines, order - keep them **byte-for-byte**.
    - **Do NOT change meaning, scope, or constraints**; do not invert conditions, ranges,
      or numeric bounds.
    - **Do NOT translate or switch languages.**
    - **Preserve technical literals in the description:** never noise inside backticks/
      fences, identifiers (function/class/parameter names), regexes, Big-O (e.g., `O(n
      log n)`), numbers/units, file paths, URLs, or special tokens/placeholders like
      `<...>`, `{...}`, `$...`.
    - **Keep the task recoverable:** include at least **one clean, unambiguous full
      sentence** stating the task, and at least **one clean mention** of each critical
      concept (e.g., "IP address", "remove leading zeroes"). No contradictions.
    - **Length:** keep the description within **30%** of its original length.
    - **Garbage & repetition limits:** each symbol-run \leq10 chars; \leq2 runs and **\leq3
      ** repetition spans per paragraph; overall result must be readable.
    - **Grammar safety:** only minor slips (articles/agreements/punctuation); never alter
      logical polarity (negations, comparatives) or numeric conditions.

    Return exactly:
    <answer>
    <noised description here><the original code stubs, unchanged>
    </answer>

    Original prompt (`original_prompt`):
    {{original_prompt}}
        """
```