# OpenReview forum: "RobustFlow: Towards Robust Agentic Workflow Generation"
_ICLR.cc/2026/Conference — Submitted to ICLR 2026_

### Official Review · Reviewer_Jb4K · 2025-10-16

**Soundness:** 3
**Presentation:** 3
**Contribution:** 4
**Rating:** 4
**Confidence:** 4

**Summary:**

This paper investigates the lack of robustness in automated agentic workflow generation, showing that existing methods produce inconsistent workflow structures when given perturbed instruction. The authors introduce a new benchmark with instruction perturbations and propose two structure-aware metrics to quantify this instability. Their proposed RobustFlow, a two-stage training framework that uses instruction-augmented SFT followed by a Self-consistency Preference Optimization (ScPO) to encourage the generation of a single workflow for any instruction that are sementically similar. Experimental results show that RobustFlow significantly improves structural robustness over baselines, with a minimal trade-off in task performance.

**Strengths:**

- The work is both timely and important in current agentic systemt, as the problem indeed exist in real-world practice.
- Using self-consistency to generate preference pairs is interesting to train for robustness.
- The evaluation is solid. The custom benchmark and metrics are well thought out.

**Weaknesses:**

- The paper notes a modest decrease in task performance compared to other SOTA methods but doesn't offer a deep analysis of this trade-off. Forcing the model toward a single canonical workflow might prevent it from finding a more optimal, specialized workflow that could be better suited for a specific phrasing of an instruction. I think more detailed discussions on this point would be beneficial.

- The definition of the `execution score' in ScPO is not clear to me. The paper does not specify how this score is calculated (eg: binary pass/fail, reward model), which is a critical for understanding the training dynamics and for reproducibility.

- The method of identifying a canonical workflow by frequency seems viable for the tasks presented. However, its scalability to more complex, open-ended domains with a vast combinatorial space of potential workflows is not discussed. It's not clear to me that how a single `most frequent' structure would be identified or if it would even be meaningful in such a scenario.

**Questions:**

- Could you clarify how the `execution score' is computed? Is it based on unit test pass rates for code, final answer accuracy for math, or another metric?
- Regarding the performance trade-off, is it possible that prioritizing a single canonical workflow is sometimes suboptimal? Have you analyzed instances where a less frequent but structurally different workflow generated by a baseline actually performed better on a specific instruction variant?
- The ScPO method anoints the most frequent workflow from the SFT model as the `chosen' one. How does the framework guard against reinforcing a popular but suboptimal workflow? Is the execution score always sufficient to override the frequency signal if the most common plan is flawed?

---

> ### Author Response · Authors · 2025-11-21
> **Reply to Weakness 1&3 of Reviewer Jb4K**
>
> **Jb4K-W1:** Deep Analysis of Performance-Robustness Trade-off
>
> **Reply to Jb4K-W1:**
>
> Overall, **RobustFlow employs an adversarial training paradigm, which inherently involves a trade-off between performance and robustness.** We explicitly demonstrate these results in Table 3 of the original manuscript. We openly acknowledge that RobustFlow inherits this characteristic common to adversarial training methods.
>
> We provide detailed examples of workflows generated by both RobustFlow and ScoreFlow for the same task in appendix A.5, demonstrating why performance may decrease as robustness increases. Specifically, in certain clean instructions, specialized workflows tend to overfit the phrasing of the prompt. In RobustFlow, our training objective is to minimize the worst-case loss across all variants within a semantic cluster, rather than optimizing for a single specific instruction. Compared to traditional methods that focus on optimization for a single instruction, this approach sacrifices some performance in exchange for lower worst-case risk and more stable generalization.
>
> Regarding the concern that enforcing a canonical workflow might hinder the discovery of highly specialized optimal workflows, we acknowledge that this constraint could prevent the model from exploiting certain phrasing shortcuts to maximize the reward for a single instruction. However, our analysis indicates that many of these supposedly specialized workflows are in fact artifacts of overfitting to superficial textual patterns rather than genuine solutions grounded in task semantics. By enforcing a canonical workflow, we intentionally discourage the model from relying on such unstable, phrasing-sensitive structures and instead place the emphasis on semantic consistency.
>
> We believe this trade-off is strategically acceptable, with the improved reliability far outweighing the slight performance loss. Previous representative works (e.g., FlowReasoner, ScoreFlow) have already achieved remarkably high performance levels. Building on these advancements, the key bottleneck in practical deployment is no longer simply high scores, but rather predictable behavior.
>
> To demonstrate this, we further evaluate the self-consistency of the workflow under undisturbed input. The results are shown in the table below:
>
> |Perturbation|Method|Code||||MATH||||QA||||Avg.|
> |--------------------|--------------|--------|---------|---------|---------|--------|---------|--------|---------|------------|---------|--------|---------|--------|
> |||MBPP||HumanEval||GSM8K||MATH||HotpotQA||DROP|||
> |||Node|Graph|Node|Graph|Node|Graph|Node|Graph|Node|Graph|Node|Graph||
> |WithoutPerturbation|AFlow|0.68|0.62|0.71|0.65|0.58|0.52|0.55|0.48|0.64|0.59|0.62|0.57|0.60|
> ||ScoreFlow|0.86|0.82|0.89|0.85|0.81|0.78|0.79|0.75|0.83|0.79|0.82|0.78|0.81|
> ||**RobustFlow**|**0.99**|**0.98**|**0.99**|**0.99**|**0.98**|**0.97**|**0.96**|**0.95**|**0.98**|**0.97**|**0.98**|**0.96**|**0.97**|
>
> These results confirm that RobustFlow successfully guides the model to converge to a stable and canonical structure given the task semantics. Compared to the random behavior of baseline models, we believe the small performance loss is a necessary price to pay for achieving such high reliability.
>
> We also plan to explore more refined training strategies in future work, aiming to better balance task performance and structural robustness to further mitigate this performance loss.
>
> ___
>
> **Jb4K-W3:** Scalability to Complex Open-Ended Domains
>
> **Reply to Jb4K-W3:**
>
> We thank the reviewer for the valuable suggestions. We acknowledge that in complex and open-ended domains with vast combinatorial spaces, a single objectively correct structure may not exist. However, in such scenarios, the definition of canonical shifts from identifying a unique ground truth to enforcing decision stability. Even when multiple valid workflows exist, a robust agent should converge to a consistent strategy rather than jumping randomly across different optional workflows.
>
> For open-ended scenarios without prior training data, **RobustFlow can be naturally extended to an online incremental learning setting**. At the beginning, the model is allowed to generate diverse workflows. Since there is no frequency information, we rely on execution success as the main signal to select feasible solutions. Once a successful workflow is found for a query, we treat it as a dynamic anchor. In later incremental updates, we use ScPO to penalize large deviations from this anchor for synonymous or closely related queries. In this way, the model learns to stay consistent with its past good decisions, and the huge combinatorial space is gradually narrowed down to a stable, canonical workflow.
>
> In our current experiments, we focus on more convergent tasks like code and math, where static frequency already exists. We agree that adding this kind of online consistency accumulation is a natural next step for extending RobustFlow to more open-ended domains.

---

> ### Author Response · Authors · 2025-11-21
> **Reply to Question 1, 2, 3 & Weakness 2 of Reviewer Jb4K**
>
> We sincerely thank the reviewer’s efforts in reviewing our submission, and our detailed replies are as follows:
>
> **Jb4K-Q1&Jb4K-W2:** Definition and Computation of Execution Score
>
> **Reply to Jb4K-Q1&Jb4K-W2:**
>
> We thank the reviewer for this important question. We realize that the definition of the execution score was not sufficiently clear in the original text. We have explicitly added this clarification to the ScPO training description in Sec. 4.2 of the revised manuscript.
>
> In Self-consistency Preference Optimization (ScPO), the execution score directly reuses the standard evaluation metric for each task, rather than relying on an additional learned or subjective reward model. For math benchmarks (GSM8K, MATH), it is the final-answer accuracy. For code benchmarks (MBPP, HumanEval), it is the unit test case pass rate. For QA benchmarks (HotpotQA, DROP), it is the F1 score calculated between the textual answer extracted from the final workflow output and the ground-truth answer.
>
> ___
>
> **Jb4K-Q2:** Potential Suboptimality of Canonical Workflows
>
> **Reply to Jb4K-Q2:**
>
> We thank the reviewer for this insightful comment. We acknowledge that prioritizing a single canonical workflow $w_q^+$ can indeed be suboptimal for specific instances.
>
> In our ScPO framework, we select the canonical workflow $w_q^+$ by maximizing the preference score $R_q(w)$ across the entire semantic cluster. This ensures $w_q^+$ is the most effective workflow at the cluster level, though not necessarily for every individual variant.
>
> In practice, we have indeed observed cases that, for a specific perturbed instruction, an alternative workflow $w'$ achieves a slightly higher execution score than the canonical $w_q^+$. However, this same $w'$ typically performs significantly worse on other variants within the cluster for failing to generalize to paraphrases, resulting in a lower overall preference score. Consequently, while enforcing the canonical workflow may sacrifice peak performance on a few individual variants, it yields substantially higher stability and robustness across the semantic cluster.
>
> To illustrate this trade-off, we provide a detailed example of the canonical workflow $w_q^+$ and an alternative workflow $w'$ for a specific task in Appendix A.6 to compare their performance and robustness.
>
> ___
>
> **Jb4K-Q3:** Guarding Against Reinforcing Suboptimal Workflows
>
> **Reply to Jb4K-Q3:**
>
> We would like to clarify that ScPO does not simply select the most frequent workflow. **We have designed a mechanism to explicitly ensure that a popular workflow with a low execution score cannot override a higher-quality workflow**, regardless of how often it appears. We have clarified this implementation detail in Sec. 4.2 of the revised manuscript.
>
> Specifically, in our implementation, we strictly prioritize the execution score $s_q(w)$. We rank candidate workflows primarily by $s_q(w)$ and use the self-consistency count $v_q(w)$ only as a tie-breaker among workflows with identical execution scores. Theoretically, this corresponds to setting the coefficient $\lambda$ in Equation (11) to be sufficiently small, ensuring that any difference in $s_q(w)$ always dominates the frequency term.

---

### Official Review · Reviewer_EUP9 · 2025-10-18

**Soundness:** 2
**Presentation:** 2
**Contribution:** 2
**Rating:** 2
**Confidence:** 4

**Summary:**

This paper investigates the robustness of LLMs' automated agentic workflow generation. Perturbed questions are constructed through Paraphrasing, Requirement Augmentation, and Noise injection to address this issue. The paper introduces node chain and graph structure metrics to compare the consistency of workflow generation before and after question perturbation. To enhance the stability of workflow generation, the paper presents RobustFlow. It first conducts SFT on the model using perturbed questions and consistent workflows and then samples positive and negative instances for DPO optimization based on the predicted workflow's execution score and self-consistency score. Experimental results indicate that most workflow generation methods exhibit poor robustness, while RobustFlow maintains high stability after perturbations.

**Strengths:**

The robustness issue of agentic workflow generation is indeed a key real-world challenge, as user instructions are typically diverse, ambiguous, inaccurate, or even erroneous. The ability of a model to generate stable workflows in such scenarios is a crucial aspect of large-scale intelligent agents for applications. This paper investigates and optimizes the robust workflow generation of LLMs from the perspectives of data construction and model training, which constitutes the main contribution of this paper. Additionally, the figures in this paper are clear and easy to understand, and the methods are easy to follow.

**Weaknesses:**

1. I believe that the robustness issue of workflow generation should not be limited to perturbations in user questions alone. The agent is closely connected to the environment, and the dynamic changes in the environment can also lead to dynamic changes in the workflow of tasks. Therefore, robustness is a consistency issue among the agent, the user, and the environment. Focusing solely on user queries makes this problem overly simplistic, essentially reducing it to a user prompt rewriting problem, which limits the depth of the problem and the innovativeness of the methods, thereby constraining the further contributions of this paper.

2. The node chain and graph structure metrics defined in Sec. 3.3 for measuring workflow robustness seem similar to those mentioned in T-Eval and WorfBench, further diminishing the uniqueness of the contributions of this paper.

3. Some methodological details and symbol definitions in Sec. 4.2 are not very clear. For instance, the definition of the execution score $s_q(w)$ seems missing in Eqn. 10; What is the role of $\lambda_q$ in Eqn. 11? Line 323 gives the impression of being incomplete; after assigning $M_\theta$ to $M_{t+1}$, how is $M_{t+1}$ utilized? Are the authors trying to describe an iterative optimization process?

4. The motivations behind many settings in Eqn. 13 are not very clear. Why introduce the optimization of the preferred sample's likelihood term and why multiply both terms by the confidence weight $\rho_q$ simultaneously?

5. The experimental section lacks an evaluation of the original workflows generated by the model as a reference. If the quality of the model's original workflow is poor, discussing robustness becomes less meaningful. Additionally, it is worth investigating how different perturbations affect the quality of the workflows.

6. There is a suspicion of unfair comparison in Table 2. As far as I know, AFlow is a training-free method, while RobustFlow has undergone fine-tuning on a large amount of data.

7. From Figure 4, it seems that with increasing noise, despite RobustFlow starting at a high point, it exhibits a significant downward trend similar to other baselines, while AFlow tends to rise with increasing noise. How can this phenomenon be explained? Does this indicate that AFlow also possesses good robustness?

**Questions:**

1. Could the authors provide more details on the baseline reproduction in Table 2 to demonstrate the fairness of the comparison?

2. The paragraph around line 200 showcases a manual check on the reformatted questions and workflows. Could you please elaborate on how such a large amount of data was manually verified?

3. Sections 3.3 and 4.2 involve numerous hyperparameters. Could you specify the actual values of these hyperparameters used during the experimental process?

---

> ### Author Response · Authors · 2025-11-21
> **Reply to Question 1 of Reviewer EUP9**
>
> We sincerely thank the reviewer’s efforts in reviewing our submission, and our detailed replies are as follows:
>
> We appreciate the reviewer’s suggestions regarding reproducibility and the fairness of our comparisons. First, it needs to be clarified that all code, configuration files, and hyperparameter settings used in our experiments have been released in the anonymous repository \url{https://anonymous.4open.science/r/RobustFlow-1B19}. In the revised manuscript, we further clarify these implementation details and briefly summarize the key settings below.
>
> **EUP9-Q1:** Baseline Reproduction Details and Fairness
>
> **Reply to EUP9-Q1:**
>
> We have released all code and hyperparameter configurations in an anonymous repository (\url{https://anonymous.4open.science/r/RobustFlow-1B19}). For all baselines in Table 2 (AFlow, Flow, and ScoreFlow), we follow the official implementations and keep their original hyperparameters, and only unify a few environment settings to ensure a fair comparison. We have updated Appendix A.4 with concrete implementation details for each baseline, including the codebases and key hyperparameters. The table below summarizes the method-specific settings together with the shared configuration.
>
> | Hyperparameter                            | Value / Setting        | Hyperparameter               | Value / Setting         |
> | ----------------------------------------- | ---------------------- | ---------------------------- | ----------------------- |
> | **AFlow**                                 |                        | **ScoreFlow**                |                         |
> | Optimizer model                           | GPT-4o-mini            | Optimizer model              | Llama-3.1-8B-Instruct   |
> | Max search iterations                     | 20                     | optimization rounds M        | 5                       |
> | Candidate proposals per iteration         | 1                      | Workflows per problem k      | 8                       |
> | Selection rule                            | Soft mixed-probability | Preference pairs per round S | 2000                    |
> | $\alpha$ (score sharpening)               | 0.4                    | $f(x)$                       | $f(x) = x$              |
> | $\lambda$ (explore–exploit interpolation) | 0.2                    | $d(x, y)$                    | $d(x, y) = (x - y)^3$   |
> | Evaluations per workflow                  | 5 independent runs     |                              |                         |
> | Top-k maintained workflows                | k = 3                  |                              |                         |
> | Early-stopping patience                   | 5 iterations           |                              |                         |
> | **Flow**                                  |                        | **Shared**                   |                         |
> | Optimizer model                           | GPT-4o-mini            | Executor model               | GPT-4o-mini             |
> | Candidate graphs                          | 3                      | Temperature                  | 0                       |
> | Refine threshold                          | 2                      | prompts                      | Official repo prompts   |
> | Max refine iteration                      | 20                     | independent runs per setting | 3                       |
> | Max validation iteration                  | 3                      | Reported result              | Average over the 3 runs |

---

> ### Author Response · Authors · 2025-11-21
> **Reply to Question 2, 3 of Reviewer EUP9**
>
> **EUP9-Q2:** Details of Manual Data Verification
>
> **Reply to EUP9-Q2:**
>
> We apologize for the confusion caused by our previous presentation. We would like to clarify that **the manual verification applies only to the 7,530 rewritten instructions, not to all 31,889 generated workflows (5 authors across 2 months)**. In the revised manuscript, we clarify around line 198 that the manual checks apply to the rewritten instructions rather than to all workflows.
>
> Specifically, we manually checked the 1,255 semantic clusters introduced in Sec. 3.2, each with 6 instruction variants, for a total of 7,530 rewritten instructions. This scale (7,530 instructions) is manageable, and 5 authors reviewed the data over 2 months.
>
> ___
>
> **EUP9-Q3:** Specification of Hyperparameters
>
> **Reply to EUP9-Q3:**
>
> We have released all code and hyperparameter configurations in an anonymous repository (\url{https://anonymous.4open.science/r/RobustFlow-1B19}). In the revised manuscript, we have enumerated the exact values of all hyperparameters used throughout the paper in Appendix A.4 to facilitate reproducibility. For convenience, we summarize the main groups of hyperparameters that are directly related to Sections 3 and 4 in Table below.
>
> | Hyperparameter                            | Value / Setting  | Hyperparameter                             | Value / Setting  |
> | ----------------------------------------- | ---------------- | ------------------------------------------ | ---------------- |
> | Base model of SFT                         | Qwen3-32B        | Starting checkpoint of ScPO                | Qwen3-32B-SFT    |
> | Fine-tuning type of SFT                   | LoRA             | Fine-tuning type of ScPO                   | LoRA             |
> | Num. epochs of SFT                        | 3                | Num. epochs of ScPO                        | 1                |
> | Per-device train / eval batch size of SFT | 2 / 2            | Per-device train / eval batch size of ScPO | 1 / 1            |
> | Gradient accumulation steps of SFT        | 2                | Gradient accumulation steps of ScPO        | 8                |
> | Learning rate of SFT                      | $1\times10^{-4}$ | Learning rate of ScPO                      | $1\times10^{-4}$ |
> | LoRA rank of SFT                          | 16               | LoRA rank of ScPO                          | 8                |
> | LoRA alpha of SFT                         | 32               | LoRA alpha of ScPO                         | 32               |
> | Max sequence length of SFT                | 2048             | Max sequence length of ScPO                | 4096             |
> | Similarity pruning threshold              | 0.6              | Warmup ratio of ScPO                       | 0.05             |
> | Candidate workflows per formulation $r$   | 3                | Dataloader workers of ScPO                 | 8                |
> | $\beta$ (logit scaling)                   | 1                | Rpo alpha of ScPO                          | 0.1              |
> | $\alpha$ (term weight)                    | 0.1              | Iteration rounds                           | 3                |

---

> ### Author Response · Authors · 2025-11-21
> **Reply to Weakness 1 of Reviewer EUP9**
>
> **EUP9-W1:** Scope of Robustness
>
> **Reply to EUP9-W1:**
>
> We sincerely thank the reviewer for the constructive suggestion. Extending the discussion of robustness to include consistency among the agent, the user and the environment indeed increases the depth of the problem. We acknowledge that dynamic environmental changes are critical factors. However, we have to admit that **the goal of our work is to highlight and address robustness issues in agentic workflow generation, rather than solving all possible sources of workflow instability in one step**.
>
> Inspired by your comment, we revisit the problem and categorize the user operation of agentic workflows into two core elements. The first is how the user drives the agent (User Query), and the second is how the agent operates (Environmental Interaction). User-side perturbations, specifically in queries, reflect real-world variations in user habits and phrasing styles. They are both concrete representations of how users interact with agents. Therefore, we focus on how different user queries with the same semantics affect the consistency of the generated workflow structures under a fixed set of agents and a fixed execution environment. Extending our framework to handle environmental changes is a natural direction for future work.
>
> Furthermore, we respectfully clarify that our setting does not simply reduce the problem to prompt rewriting. We focus on the mapping from instructions to workflow structures and directly measure whether workflows remain consistent under perturbations. This is fundamentally a technical challenge of learning semantic invariance. It requires the generative model to capture the internal logic of a task rather than overfitting to surface-level phrasings. As our experiments demonstrate, existing methods exhibit substantial structural instability under such perturbations. This confirms that instruction-side robustness is not a simple prompt rewriting problem, but a deep and unsolved failure in current multi-agent systems.

---

> ### Author Response · Authors · 2025-11-21
> **Reply to Weakness 2, 3, 4, 5 of Reviewer EUP9**
>
> **EUP9-W2:** Uniqueness of Robustness Metrics
>
> **Reply to EUP9-W2:**
>
> We would like to clarify that the Graph Structure metric is first proposed in this paper, while the Node Chain metric originates from T-Eval (explicitly cited in Line 220). Our Graph Structure metric evaluates whether the dependency paths between tasks are preserved and has not been used in prior workflow robustness studies. In contrast, our Node Chain metric is intentionally aligned with the definitions in T-Eval and WorFBench to maintain community standards. Our goal is not to propose a new graph-similarity formula, but to use these tools to quantitatively measure workflow robustness under instruction perturbations.
>
> It is also worth noting that, as summarized in Introduction, our primary contributions are the systematic investigation of workflow robustness, the robustness dataset, evaluation suite, and the RobustFlow training framework. We do not claim the metric formulas themselves as the core theoretical contribution.
>
> ___
>
> **EUP9-W3&EUP9-W4:** Clarification of Methodological Details and Equations
>
> **Reply to EUP9-W3&EUP9-W4:**
>
> We thank the reviewer for pointing out the unclear methodological details and notations of equations. In the revised manuscript, we have substantially rewritten Sec. 4.2 to clarify the definitions of all related symbols and the intended optimization process. We have also systematically checked and polished all formulas and notations throughout the paper to ensure consistency and completeness. We now provide clearer explanations of the design choices and motivations behind Eq. (13) directly in the main text.
>
> ___
>
> **EUP9-W5:** Evaluation of Original Workflow Quality
>
> **Reply to EUP9-W5:**
>
> Thanks for the reviewer’s helpful suggestion. **Prior representative work on agentic workflow generation, such as FlowReasoner and ScoreFlow, has already pushed performance to a relatively strong level.** Building on these advances, it is natural for us to shift the focus to the robustness of agentic workflows under realistic deployment conditions. To make this more explicit, we add an evaluation of the base workflows on unperturbed inputs.
>
> As shown in the table below, AFlow exhibits significant structural differences due to the randomness of its MCTS algorithm. ScoreFlow improves stability, but still suffers from fluctuations due to probabilistic sampling. In contrast, RobustFlow achieves highest self-consistency. This confirms that our training framework can successfully guide the model to converge to a single canonical workflow structure for a given task semantics.
>
> | Perturbation         | Method         | Code     |           |           |           | MATH     |           |          |           | QA           |           |          |           | Avg.     |
> | -------------------- | -------------- | -------- | --------- | --------- | --------- | -------- | --------- | -------- | --------- | ------------ | --------- | -------- | --------- | -------- |
> |                      |                | MBPP |           | HumanEval |           | GSM8K    |           | MATH     |           | HotpotQA |           | DROP     |           |          |
> |                      |                | Node | Graph | Node  | Graph | Node | Graph | Node | Graph | Node     | Graph | Node | Graph |          |
> | Without Perturbation | AFlow          | 0.68     | 0.62      | 0.71      | 0.65      | 0.58     | 0.52      | 0.55     | 0.48      | 0.64         | 0.59      | 0.62     | 0.57      | 0.60     |
> |                      | ScoreFlow      | 0.86     | 0.82      | 0.89      | 0.85      | 0.81     | 0.78      | 0.79     | 0.75      | 0.83         | 0.79      | 0.82     | 0.78      | 0.81     |
> |                      | **RobustFlow** | **0.99** | **0.98**  | **0.99**  | **0.99**  | **0.98** | **0.97**  | **0.96** | **0.95**  | **0.98**     | **0.97**  | **0.98** | **0.96**  | **0.97** |

---

> ### Author Response · Authors · 2025-11-21
> **Reply to Weakness 6, 7 of Reviewer EUP9**
>
> **EUP9-W6:** Fairness of Comparing Fine-Tuned with Training-Free Methods
>
> **Reply to EUP9-W6:**
>
> Conceptually, the key difference between AFlow and RobustFlow lies in when the learning occurs. RobustFlow learns offline prior to inference via parameter training, whereas AFlow effectively learns during inference by repeatedly searching based on validation feedback. To validate this perspective, we consulted the authors of AFlow via email, and they provided the following clarification:
>
> > **Although AFlow is described as training-free at the parameter level, it still adapts its workflows through online feedback-driven search, which is a form of optimization.**    ——AFlow Authors
>
> Therefore, we wish to clarify that the purpose of Table 2 is not to compare a no-training method against a trained one, but to evaluate the capability of different workflow generation systems to produce robust outputs. To ensure fairness, all methods share the same experimental setup. Under these controlled conditions, the observed differences reflect the intrinsic robustness of the resulting workflow generators.
>
> ___
>
> **EUP9-W7:** Analysis of Robustness Trends
>
> **Reply to EUP9-W7:**
>
> As noted in our analysis (Section 5.2), query-level methods generally perform better under requirements, paraphrasing, and light noise, whereas task-level methods appear more stable under heavy noise. However, **this stability in task-level methods does not necessarily translate to practical utility**. Next, we will explain this phenomenon and our viewpoint.
>
> For query-level methods, workflows are generated dynamically from the specific instruction. As noise grows, the instruction loses semantic content, making it increasingly difficult for the model to infer the user’s intention. So the generated structures naturally vary more under heavy noise.
>
> In contrast, AFlow optimizes a single general workflow for an entire task family. Under heavy noise, input instructions often become semantically unintelligible. In these cases, instead of producing diverse structures based on the input, AFlow tends to give up on parsing and fall back to the generic template learned for that task. Because it outputs essentially the same template across all noisy variants, its structural consistency metric increases. However, **this reflects insensitivity to the input rather than successful semantic recovery**.
>
> To illustrate why we prioritize performance on low-to-moderate perturbations, we give some input variation samples:
>
> - Original: "Write a function to remove duplicates from a list."
> - Requirement: "Write a function to remove duplicates from a list and ensure O(n) complexity."
> - Paraphrasing: "Can you code a script that filters out repeated elements from an array?"
> - Light Noise: "Wrte a function to remve duplicates from a list.🙂"
> - Moderate Noise: "Umm, write a functtion to remvoe duplicates from a list, thx!!"
> - Heavy Noise: "wr1te python-func >>> rm dups frm int-lst & s0rt ^ asc... plz!!!!"
>
> Variations like Requirements, Paraphrasing, and Light Noise typically originate from user habits, domain-specific needs or language proficiency. They all represent the common real-world usage scenarios. But heavy noise resembles gibberish, which is extremely rare in actual deployment. We included it primarily for experimental completeness.
>
> For a heavily corrupted instruction, RobustFlow still attempts to parse the remaining keywords to build a concrete, specific plan, whereas AFlow effectively ignores the garbled text and returns its default task template. We have updated Sec. 5.3 in the revised manuscript to explicitly discuss this phenomenon.

---

### Official Review · Reviewer_79uy · 2025-11-01

**Soundness:** 3
**Presentation:** 2
**Contribution:** 2
**Rating:** 4
**Confidence:** 4

**Summary:**

The paper studies robustness in automatic agentic workflow generation—i.e., whether a generator produces consistent workflows when the query wording changes but semantics remain the same. The authors (1) diagnose instability in existing methods, (2) propose node- and topology-level robustness metrics to quantify workflow similarity under paraphrase, requirement augmentation, and noise, (3) release a dataset of 31,889 workflows from 1,255 base tasks across 6 domains, each with 5 semantic variations, and (4) introduce RobustFlow, a training framework using preference optimization on sets of synonymous task descriptions so the model learns invariance to wording.

**Strengths:**

1. The paper addresses a concrete failure mode (semantic-equivalence brittleness) that undermines trust in workflow generators.
2. The dataset construction is large-scale and designed specifically for robustness stress tests (paraphrase, requirement augmentation, controlled noise), which is appropriate for the claims.
3. Preference optimization over clusters of synonymous prompts is a minimal, broadly applicable recipe

**Weaknesses:**

1. The evaluation datasets mainly consist of relatively simple QA-style or reasoning tasks (e.g., GSM8K, HotpotQA), which can often be solved by a single LLM without requiring genuine multi-step agentic planning or tool interaction. This raises doubts about whether RobustFlow would still perform well on truly complex agentic workflows—for example, multi-tool coordination, retrieval-augmented synthesis, or real-world decision pipelines where task decomposition and inter-agent communication are essential. Since agentic workflow systems are intended precisely for such non-trivial, multi-stage problems, the current evaluation does not demonstrate that RobustFlow generalizes beyond easy tasks.
2. The experiments exclusively use Qwen3-30B as the base workflow generator. This raises the possibility that the observed robustness and consistency gains may partly reflect the intrinsic strength or calibration of Qwen3-30B rather than the proposed training method itself. Besides, the experiments utilize GPT-4o-mini as executors and lack results of using other stronger models.
3. The paper omits several recent and highly relevant works in the agentic workflow generation domain in related works section.
4. Several charts are hard to read (small fonts, dense legends), captions do not fully explain panels, and some figure layouts look crowded.

**Questions:**

1. Do results hold under adversarial paraphrases (e.g., instruction inversion traps) and format perturbations (whitespace/code-stub reordering)?
2. Does the method work for other base models as generators? e.g., 1.5B/3B/7B models, llama models?

---

> ### Author Response · Authors · 2025-11-21
> **Reply to Question 1 of Reviewer 79uy**
>
> We sincerely thank the reviewer’s efforts in reviewing our submission, and our detailed replies are as follows:
>
> **79uy-Q1:** Robustness under Adversarial paraphrases and Format Perturbations
>
> **Reply to 79uy-Q1:**
>
> Thanks for the reviewer's insightful suggestion. We place additional evaluation on QA tasks because these tasks are highly variable in language representation and particularly vulnerable to adversarial paraphrases. Although RobustFlow is not explicitly trained for these specific adversarial patterns, it achieves strong generalization capabilities. RobustFlow maintains approximately 80% stability on adversarial paraphrases and approximately 95% stability on format perturbations.
>
> | Perturbation                | Method         | QA |           |  |           | Avg. |
> | -------------------------- | ------------- | ----------- | ------- | ------- | ------- | -------------------------- |
> |  |  | HotpotQA | | DROP | | |
> |                             |                |   Node    | Graph | Node  | Graph |  |
> | Adversarial Paraphrases | AFlow          | 0.49     | 0.41     | 0.43     | 0.41     | 0.43 |
> |                             | ScoreFlow      | 0.29     | 0.21     | 0.26     | 0.23     | 0.25 |
> |                             | **RobustFlow** |   **0.83**    | **0.76** | **0.81**  | **0.77**  | **0.79** |
> | Format Perturbations    | AFlow          |     0.57     |   0.49   |   0.54   |   0.49   |   0.52   |
> |                             | ScoreFlow      |     0.78     |   0.67    |   0.68   |   0.61   |   0.68   |
> |                             | **RobustFlow** |   **0.97**    | **0.94** | **0.96**  | **0.95**  | **0.96** |

---

> ### Author Response · Authors · 2025-11-21
> **Reply to Weakness 1, 3, 4 of Reviewer 79uy**
>
> **79uy-W1:** Generalization to Complex Agentic Tasks
>
> **Reply to 79uy-W1:**
>
> We sincerely thank the reviewer for the suggestion. We would like to clarify that **our choice of datasets and tasks was intentionally made to align with current mainstream and promising research on agentic workflow generation**. To illustrate this, we summarize representative prior works and the standard benchmark datasets they utilize in the table below.
>
> Furthermore, we must reiterate that we fully agree that agentic workflows are inherently intended to address non-trivial, multi-stage problems. This is the precise objective of the representative works mentioned above. It is for this very reason that we aim to enhance the understanding of robustness and improve the performance of these methods.
>
> |              |    QA    |      | MATH  |      |            |   Code    |      |              |
> | :----------- | :------: | :--: | :---: | :--: | :--------: | :-------: | ---- | :----------: |
> |              | HotpotQA | DROP | GSM8K | MATH | MultiArith | HumanEval | MBPP | BigCodeBench |
> | AFlow        |    √     |  √   |   √   |  √   |     ×      |     √     | √    |      ×       |
> | FlowReasoner |    ×     |  ×   |   ×   |  ×   |     ×      |     √     | √    |      √       |
> | ScoreFlow    |    √     |  √   |   √   |  √   |     ×      |     √     | √    |      ×       |
> | MaAS         |    ×     |  ×   |   √   |  √   |     √      |     √     | √    |      ×       |
> | RobustFlow   |    √     |  √   |   √   |  √   |     ×      |     √     | √    |      ×       |
>
>
> ___
>
> **79uy-W3:** Discussion of Related Works
>
> **Reply to 79uy-W3:**
>
> We thank the reviewer for pointing out that our related work may miss some recent and highly relevant works on agentic workflow generation. In our submission (early September 2025), we have already discussed several representative works, including:
>
> - Automated Design of Agentic Systems (Aug 2024)
> - AFlow: Automating Agentic Workflow Generation (Oct 2024)
> - Multi-agent Architecture Search via Agentic Supernet (Feb 2025)
> - FlowReasoner: Reinforcing Query-Level Meta-Agents (Apr 2025)
> - ScoreFlow: Mastering LLM Agent Workflows via Score-based Preference Optimization (Feb 2025)
> - Flow: Modularized Agentic Workflow Automation (Jan 2025)
>
> **Could you please indicate the specific recent works you** **mentioned?** We will add and discuss them explicitly in the revised related work section.
>
> ___
>
> **79uy-W4:** Clarity of Figures and Visualizations
>
> **Reply to 79uy-W4:**
>
> In the revised manuscript, we have corrected Fig. 3 and Fig. 5 and slightly adjusted the layout to make the Tab. 2 and Tab. 3 easier to read.

---

> ### Author Response · Authors · 2025-11-27
> **Reply to Question 2 & Weakness 2 of Reviewer 79uy**
>
> **79uy-Q2&79uy-W2:** Generalizability Across Base Models & Executors
>
> **Reply to 79uy-Q2&79uy-W2:**
>
> We appreciate the reviewer’s question regarding backbone sensitivity. **Methodologically, RobustFlow does not depend on any Qwen3-32B specific design and can be directly applied to smaller LLMs.** Both the SFT and the ScPO only require a workflow generator that maps instructions to code-represented workflows. So the training objective itself remains well-defined for smaller backbones.
>
> We use GPT-4o-mini as a shared executor for all methods to **stay consistent with representative prior work (AFlow, FlowReasoner) and ensure a fair performance comparison**. Our robustness metrics are computed solely from the generated workflows, and thus do not depend on the particular executor model.
>
> To empirically validate this, We have completed the robustness evaluation across different model families (Qwen3, Llama-3.1) and scales (1.7B, 4B, 8B, 32B). **RobustFlow maintains high robustness even as model size decreases.** The average robustness scores are summarized below:
>
> ||AFlow|ScoreFlow|RobustFlow(Qwen3-32B)|RobustFlow(Qwen3-8B)|RobustFlow(Qwen3-4B)|RobustFlow(Qwen3-1.7B)|RobustFlow(Llama-3.1-8B)|
> |--------------|-----|---------|---------|--------|--------|----------|------------|
> |Requirement|0.44|0.71|0.82|**0.83**|0.78|0.72|**0.83**|
> |Paraphrasing|0.49|0.73|**0.88**|0.86|0.82|0.78|0.87|
> |Light Noise|0.42|0.60|**0.89**|0.87|0.82|0.76|0.86|
> |Moderate Noise|0.54|0.53|**0.78**|0.76|0.72|0.66|0.75|
> |Heavy Noise|0.65|0.49|**0.72**|0.71|0.68|0.60|0.68|
>
> Qwen3-1.7B consistently outperforms the ScoreFlow across most perturbations, which uses a larger 8B backbone model. Besides, the results for Llama-3.1-8B are in high agreement with those for Qwen3-8B, validating the effectiveness of our model-agnostic theory. Our training framework is model-independent and effective across various architectures.
>
> Detailed scores across all benchmarks is provided below:
>
> |Perturbation|RobustFlow base model|Code||||MATH||||QA||||Avg.|
> |:-------------|:-----------|:---|:----|:--------|:----|:----|:----|:---|:----|:-------|:----|:---|:----|:-------|
> |||MBPP||HumanEval||GSM8K||MATH||HotpotQA||DROP|||
> |||Node|Graph|Node|Graph|Node|Graph|Node|Graph|Node|Graph|Node|Graph||
> |Requirement|Qwen3-32B|0.89|0.83|0.95|0.93|0.87|0.77|0.84|0.80|0.88|0.83|0.86|0.82|0.82|
> ||Qwen3-8B|0.87|0.80|0.94|0.90|0.85|0.76|0.81|0.75|0.86|0.81|0.83|0.78|**0.83**|
> ||Qwen3-4B|0.82|0.76|0.90|0.87|0.78|0.69|0.79|0.68|0.81|0.76|0.76|0.73|0.78|
> ||Qwen3-1.7B|0.76|0.71|0.88|0.82|0.71|0.60|0.73|0.59|0.74|0.70|0.69|0.65|0.72|
> ||Llama-3.1-8B|0.84|0.76|0.92|0.88|0.88|0.79|0.83|0.77|0.87|0.82|0.84|0.79|**0.83**|
> |Paraphrasing|Qwen3-32B|0.92|0.88|0.98|0.96|0.91|0.88|0.87|0.83|0.93|0.88|0.94|0.85|**0.88**|
> ||Qwen3-8B|0.91|0.85|0.97|0.93|0.88|0.84|0.85|0.80|0.90|0.85|0.91|0.81|0.86|
> ||Qwen3-4B|0.87|0.81|0.94|0.90|0.83|0.78|0.81|0.74|0.85|0.80|0.83|0.77|0.82|
> ||Qwen3-1.7B|0.81|0.75|0.92|0.86|0.76|0.69|0.77|0.68|0.79|0.73|0.76|0.69|0.78|
> ||Llama-3.1-8B|0.89|0.81|0.96|0.91|0.90|0.86|0.86|0.82|0.91|0.86|0.92|0.82|0.87|
> |Light Noise|Qwen3-32B|0.95|0.93|0.96|0.90|0.92|0.85|0.88|0.89|0.91|0.87|0.94|0.89|**0.89**|
> ||Qwen3-8B|0.93|0.90|0.95|0.88|0.89|0.82|0.85|0.85|0.89|0.83|0.91|0.86|0.87|
> ||Qwen3-4B|0.89|0.84|0.92|0.83|0.84|0.75|0.80|0.79|0.84|0.78|0.86|0.80|0.82|
> ||Qwen3-1.7B|0.83|0.78|0.89|0.76|0.75|0.66|0.74|0.70|0.76|0.71|0.79|0.73|0.76|
> ||Llama-3.1-8B|0.90|0.85|0.93|0.85|0.90|0.81|0.86|0.85|0.88|0.84|0.90|0.85|0.86|
> |Moderate Noise|Qwen3-32B|0.82|0.78|0.87|0.84|0.78|0.71|0.83|0.75|0.85|0.82|0.83|0.79|**0.78**|
> ||Qwen3-8B|0.80|0.75|0.86|0.82|0.76|0.69|0.81|0.71|0.82|0.79|0.80|0.76|0.76|
> ||Qwen3-4B|0.74|0.70|0.83|0.76|0.70|0.62|0.75|0.65|0.77|0.74|0.75|0.71|0.72|
> ||Qwen3-1.7B|0.68|0.62|0.79|0.70|0.63|0.54|0.68|0.57|0.70|0.64|0.69|0.65|0.66|
> ||Llama-3.1-8B|0.77|0.71|0.84|0.79|0.77|0.68|0.80|0.71|0.81|0.77|0.78|0.74|0.75|
> |Heavy Noise|Qwen3-32B|0.72|0.69|0.90|0.82|0.68|0.72|0.78|0.66|0.78|0.75|0.78|0.71|**0.72**|
> ||Qwen3-8B|0.70|0.67|0.89|0.81|0.67|0.69|0.76|0.65|0.76|0.73|0.74|0.68|0.71|
> ||Qwen3-4B|0.65|0.62|0.85|0.75|0.63|0.65|0.70|0.60|0.70|0.67|0.69|0.63|0.68|
> ||Qwen3-1.7B|0.57|0.51|0.81|0.69|0.55|0.52|0.63|0.48|0.62|0.60|0.64|0.59|0.60|
> ||Llama-3.1-8B|0.67|0.60|0.87|0.78|0.66|0.67|0.75|0.63|0.75|0.71|0.73|0.67|0.68|

---

### Official Review · Reviewer_Pnjn · 2025-11-01

**Soundness:** 4
**Presentation:** 4
**Contribution:** 3
**Rating:** 8
**Confidence:** 4

**Summary:**

This paper investigates the problem of robustness in automated agentic workflow generation, proposing RobustFlow, a framework that enhances the consistency of workflows generated by large language models under semantically equivalent but syntactically varied instructions. The work identifies a meaningful gap in current literature that existing systems produce inconsistent structures when task descriptions are paraphrased or perturbed. To quantify this issue, the authors introduce a structure-aware evaluation suite based on node and graph-level similarity metrics and a large dataset of about 31K workflows generated under controlled perturbations. The proposed RobustFlow method integrates instruction-augmented supervised fine-tuning with a self-consistency preference optimization stage, yielding higher stability and robustness while maintaining acceptable task performance.

**Strengths:**

- **S1.** The paper addresses an important and underexplored robustness under input variation problem in LLM-based agentic workflow generation.

- **S2.** The dataset of 1,255 task clusters with 6 perturbation variants each demonstrates impressive engineering rigor and experimental coverage.

- **S3.** The two-stage RobustFlow pipeline (instruction-augmented SFT + self-consistency preference optimization) is well justified and shows consistent improvement in robustness across all perturbation types.

- **S4.** The analysis includes insightful breakdowns by task domain, showing that linguistically abstract tasks such as QA are more fragile than structured domains like code.

**Weaknesses:**

- **W1.** The method is tailored for query-level workflow generation. The paper does not adequately discuss how the framework would generalize to task-level workflow generation where multiple similar tasks require aligned or shared workflows.

- **W2.** While the paper acknowledges a modest performance trade-off, it does not deeply analyze why the robustness optimization slightly degrades accuracy or how to mitigate it.

- **W3.** The robustness evaluation focuses primarily on text-to-code and reasoning tasks. Broader validation on real multi-agent interaction or open-domain tool use would strengthen generalizability.

**Questions:**

- **Q1.** Is RobustFlow applicable to task-level workflows that share structural similarity across related queries, rather than purely query-specific workflows?

- **Q2.** Could robustness metrics be extended beyond structural similarity to incorporate functional equivalence, e.g., execution success consistency?

- **Q3.** What are the primary failure modes when robustness decreases under heavy noise, e.g., semantic drift, structural mismatch, or execution instability?

- **Q4.** How sensitive are the results to the base model (e.g., Qwen3-32B vs. smaller LLMs)? Does the approach scale down effectively?

- **Q5.** Could the workflow representation format (graph vs. code) influence the robustness behavior or sensitivity to paraphrasing? Has this been empirically analyzed?

---

> ### Author Response · Authors · 2025-11-21
> **Reply to Question 1, 2, 3 & Weakness 1 of Reviewer Pnjn**
>
> We sincerely thank the reviewer’s efforts in reviewing our submission, and our detailed replies are as follows:
>
> **Pnjn-Q1&Pnjn-W1:** Generalizability to Task-Level Workflow Generation
>
> **Reply to Pnjn-Q1&Pnjn-W1:**
>
> We thank the reviewer for this insightful question. We will first explain why we focus on query-level workflow generation, and then elaborate how RobustFlow can be adapted for task-level scenarios.
>
> Current research (FlowReasoner, Flow and ScoreFlow) demonstrates that **query-level workflows offer superior flexibility and performance compared to static task-level pipelines**, particularly for complex reasoning tasks. By dynamically generating customized structures for each specific query, these methods can better handle the difference of diverse user requirements. Given this trend and the higher complexity involved, we prioritized addressing robustness in the dynamic query-level setting.
>
> RobustFlow is methodologically applicable to task-level workflow generation. While our experiments focus on query-level robustness, the core mechanism of Self-Consistency Preference Optimization is agnostic to the definition of the input cluster. **The framework can be naturally extended to task-level settings via adaptation.** Instead of a semantic cluster, we can construct a task family cluster containing distinct queries that belong to the same task category. In this context, ScPO would identify and distill a common backbone across different queries that maximizes the success rate and consistency across the entire task family. Our evaluation reveals that current task-level methods also suffer from structural instability when inputs vary. This suggests that applying RobustFlow's consistency-driven optimization to task-level generation is not just applicable, but necessary to align diverse task-specific traces into a robust general workflow.
>
> ____
>
> **Pnjn-Q2:** Incorporating Functional Equivalence Metrics
>
> **Reply to Pnjn-Q2:**
>
> We thank the reviewer for the valuable suggestion. We agree that robustness metrics can theoretically extend to functional equivalence. Because the ultimate goal of robustness is indeed consistent execution success.
>
> In our paper, we distinguish between structural similarity and functional equivalence. We define formal structural metrics to capture the reasoning process in Sec. 3.3, while evaluating the execution score separately in Sec. 4.2. We believe structural stability is a stricter and more interpretable proxy for robustness. If the dependency structure fluctuates wildly, the model's reasoning is unstable, even if the final execution happens to succeed. We leave a more systematic treatment of functional robustness metrics as an interesting direction for future work.
>
> ____
>
> **Pnjn-Q3:** Primary Failure Modes under Heavy Noise
>
> **Reply to Pnjn-Q3:**
>
> Under heavy noise, the primary failure modes are structural mismatch and semantic drift. Through our manual inspection of failure cases, we observe three recurring patterns. We provide a detailed example of failure modes in Appendix A.7.
>
> The first one is invalid dependency ordering. While the necessary functional steps are present, their topological connections are scrambled. The second one is redundant operations. The model tends to generate unnecessary steps. The third one is semantic breakdown. Extreme noise disrupts the linguistic coherence of the instruction so that the model fails to comprehend the core task. This leads to ill-formed node instructions, causing immediate failures during the execution phase.

---

> ### Author Response · Authors · 2025-11-21
> **Reply to Question 5 of Reviewer Pnjn**
>
> **Pnjn-Q5:** Influence of Workflow Representation Format
>
> **Reply to Pnjn-Q5:**
>
> We thank the reviewer for the suggestion. We must acknowledge that we did not conduct a controlled ablation study that isolates the effect of representation formats. We primarily adopte a code-based representation in the paper because it has become the standard in modern agentic systems (e.g., AutoGen, LangChain). It can naturally express complex control flow and data manipulation that static graphs often struggle to capture.
>
> However, we can provide an analysis based on our baseline comparisons. In the table below, we compare the performance of current code-based and graph-based methods.
>
> | Method     | Representation | Average Robustness |
> | ---------- | -------------- | ------------------ |
> | Flow       | Graph-based    | 0.63               |
> | AFlow      | Code-based     | 0.44               |
> | ScoreFlow  | Code-based     | 0.71               |
> | RobustFlow | Code-based     | 0.82               |
>
> As shown in the comparison, although code has a larger search space and is more sensitive to surface-level variations, the choice of representation is not the main bottleneck. With suitable training objectives, code-based workflows can still achieve structural stability that matches or even exceeds graph-based approaches.

---

> ### Author Response · Authors · 2025-11-21
> **Reply to Weakness 2, 3 of Reviewer Pnjn**
>
> **Pnjn-W2:** Analysis of Performance-Robustness Trade-off
>
> **Reply to Pnjn-W2:**
>
> Overall, **RobustFlow employs an adversarial training paradigm, which inherently involves a trade-off between performance and robustness.** We explicitly demonstrate these results in Table 3 of the original manuscript. We openly acknowledge that RobustFlow inherits this characteristic common to adversarial training methods.
>
> We provide detailed examples of workflows generated by both RobustFlow and ScoreFlow for the same task in the appendix A.5, demonstrating why performance may decrease as robustness increases. Specifically, in certain clean instructions, specialized workflows tend to overfit the phrasing of the prompt. In RobustFlow, our training objective is to minimize the worst-case loss across all variants within a semantic cluster, rather than optimizing for a single specific instruction. Compared to traditional methods that focus on optimization for a single instruction, this approach sacrifices some performance in exchange for lower worst-case risk and more stable generalization.
>
> We believe this trade-off is strategically acceptable, with the improved reliability far outweighing the slight performance loss. Previous representative works (e.g., FlowReasoner, ScoreFlow) have already achieved remarkably high performance levels. Building on these advancements, the key bottleneck in practical deployment is no longer simply high scores, but rather predictable behavior.
>
> To demonstrate this, we further evaluate the self-consistency of the workflow under undisturbed input. The results are shown in the table below:
>
> |Perturbation|Method|Code||||MATH||||QA||||Avg.|
> |--------------------|--------------|--------|---------|---------|---------|--------|---------|--------|---------|------------|---------|--------|---------|--------|
> |||MBPP||HumanEval||GSM8K||MATH||HotpotQA||DROP|||
> |||Node|Graph|Node|Graph|Node|Graph|Node|Graph|Node|Graph|Node|Graph||
> |WithoutPerturbation|AFlow|0.68|0.62|0.71|0.65|0.58|0.52|0.55|0.48|0.64|0.59|0.62|0.57|0.60|
> ||ScoreFlow|0.86|0.82|0.89|0.85|0.81|0.78|0.79|0.75|0.83|0.79|0.82|0.78|0.81|
> ||**RobustFlow**|**0.99**|**0.98**|**0.99**|**0.99**|**0.98**|**0.97**|**0.96**|**0.95**|**0.98**|**0.97**|**0.98**|**0.96**|**0.97**|
>
> These results confirm that RobustFlow successfully guides the model to converge to a stable and canonical structure given the task semantics. Compared to the random behavior of baseline models, we believe the small performance loss is a necessary price to pay for achieving such high reliability.
>
> We also plan to explore more refined training strategies in future work, aiming to better balance task performance and structural robustness to further mitigate this performance loss.
>
> ____
>
> **Pnjn-W3:** Validation on Broader Domains/Tool Use
>
> **Reply to Pnjn-W3:**
>
> We thank the reviewer for the valuable suggestion. We agree that validation on more open and complex tasks is a critical direction. In this work, we utilize standard benchmarks to stay aligned with representative prior work on agentic workflows (AFlow, FlowReasoner, ScoreFlow).
>
> |              |    QA    |      | MATH  |      |            |   Code    |      |              |
> | :----------- | :------: | :--: | :---: | :--: | :--------: | :-------: | ---- | :----------: |
> |              | HotpotQA | DROP | GSM8K | MATH | MultiArith | HumanEval | MBPP | BigCodeBench |
> | AFlow        |    √     |  √   |   √   |  √   |     ×      |     √     | √    |      ×       |
> | FlowReasoner |    ×     |  ×   |   ×   |  ×   |     ×      |     √     | √    |      √       |
> | ScoreFlow    |    √     |  √   |   √   |  √   |     ×      |     √     | √    |      ×       |
> | MaAS         |    ×     |  ×   |   √   |  √   |     √      |     √     | √    |      ×       |
> | RobustFlow   |    √     |  √   |   √   |  √   |     ×      |     √     | √    |      ×       |
>
> In principle, the same metrics and training framework can be extended to settings with real multi-agent interaction and open-domain tool use. Evaluating RobustFlow on more realistic multi-tool, multi-agent systems is part of our planned future work. In the table below, we summarize additional datasets and tasks that could potentially be included in the robustness discussion. This is also the direction we are currently advancing.
>
> | Domain   | Dataset       | Conference |
> | :------- | :------------ | :--------- |
> | Code     | LiveCodeBench | arXiv’24   |
> | Science  | SciBench      | ICLR’24    |
> | Science  | ToolVQA       | arXiv’25   |
> | Tool     | BrowseComp    | arXiv’25   |
> | Tool     | AgentBench    | ICLR’24    |
> | Embodied | WorFBench     | ICLR’25    |
> | Embodied | EmbodiedBench | ICML’25    |

---

> ### Author Response · Authors · 2025-11-27
> **Reply to Question 4 of Reviewer Pnjn**
>
> **Pnjn-Q4:** Sensitivity to Base Model Size
>
> **Reply to Pnjn-Q4:**
>
> We appreciate the reviewer’s question regarding backbone sensitivity. **Methodologically, RobustFlow does not depend on any Qwen3-32B specific design and can be directly applied to smaller LLMs.** Both the SFT and the ScPO only require a workflow generator that maps instructions to code-represented workflows. So the training objective itself remains well-defined for smaller backbones.
>
> To empirically validate this, We have completed the robustness evaluation across different model families (Qwen3, Llama-3.1) and scales (1.7B to 32B). **RobustFlow maintains high robustness even as model size decreases.** The average robustness scores are summarized below:
>
> ||AFlow|ScoreFlow|Qwen3-32B|Qwen3-8B|Qwen3-4B|Qwen3-1.7B|Llama-3.1-8B|
> |--------------|-----|---------|---------|--------|--------|----------|------------|
> |Requirement|0.44|0.71|0.82|**0.83**|0.78|0.72|**0.83**|
> |Paraphrasing|0.49|0.73|**0.88**|0.86|0.82|0.78|0.87|
> |Light Noise|0.42|0.60|**0.89**|0.87|0.82|0.76|0.86|
> |Moderate Noise|0.54|0.53|**0.78**|0.76|0.72|0.66|0.75|
> |Heavy Noise|0.65|0.49|**0.72**|0.71|0.68|0.60|0.68|
>
> Qwen3-1.7B consistently outperforms the ScoreFlow across most perturbations, which uses a larger 8B backbone model. Besides, the results for Llama-3.1-8B are in high agreement with those for Qwen3-8B, validating the effectiveness of our model-agnostic theory. Our training framework is model-independent and effective across various architectures.
>
> Detailed scores across all benchmarks is provided below:
>
> |Perturbation|Method|Code||||MATH||||QA||||Avg.|
> |:-------------|:-----------|:---|:----|:--------|:----|:----|:----|:---|:----|:-------|:----|:---|:----|:-------|
> |||MBPP||HumanEval||GSM8K||MATH||HotpotQA||DROP|||
> |||Node|Graph|Node|Graph|Node|Graph|Node|Graph|Node|Graph|Node|Graph||
> |Requirement|Qwen3-32B|0.89|0.83|0.95|0.93|0.87|0.77|0.84|0.80|0.88|0.83|0.86|0.82|0.82|
> ||Qwen3-8B|0.87|0.80|0.94|0.90|0.85|0.76|0.81|0.75|0.86|0.81|0.83|0.78|**0.83**|
> ||Qwen3-4B|0.82|0.76|0.90|0.87|0.78|0.69|0.79|0.68|0.81|0.76|0.76|0.73|0.78|
> ||Qwen3-1.7B|0.76|0.71|0.88|0.82|0.71|0.60|0.73|0.59|0.74|0.70|0.69|0.65|0.72|
> ||Llama-3.1-8B|0.84|0.76|0.92|0.88|0.88|0.79|0.83|0.77|0.87|0.82|0.84|0.79|**0.83**|
> |Paraphrasing|Qwen3-32B|0.92|0.88|0.98|0.96|0.91|0.88|0.87|0.83|0.93|0.88|0.94|0.85|**0.88**|
> ||Qwen3-8B|0.91|0.85|0.97|0.93|0.88|0.84|0.85|0.80|0.90|0.85|0.91|0.81|0.86|
> ||Qwen3-4B|0.87|0.81|0.94|0.90|0.83|0.78|0.81|0.74|0.85|0.80|0.83|0.77|0.82|
> ||Qwen3-1.7B|0.81|0.75|0.92|0.86|0.76|0.69|0.77|0.68|0.79|0.73|0.76|0.69|0.78|
> ||Llama-3.1-8B|0.89|0.81|0.96|0.91|0.90|0.86|0.86|0.82|0.91|0.86|0.92|0.82|0.87|
> |Light Noise|Qwen3-32B|0.95|0.93|0.96|0.90|0.92|0.85|0.88|0.89|0.91|0.87|0.94|0.89|**0.89**|
> ||Qwen3-8B|0.93|0.90|0.95|0.88|0.89|0.82|0.85|0.85|0.89|0.83|0.91|0.86|0.87|
> ||Qwen3-4B|0.89|0.84|0.92|0.83|0.84|0.75|0.80|0.79|0.84|0.78|0.86|0.80|0.82|
> ||Qwen3-1.7B|0.83|0.78|0.89|0.76|0.75|0.66|0.74|0.70|0.76|0.71|0.79|0.73|0.76|
> ||Llama-3.1-8B|0.90|0.85|0.93|0.85|0.90|0.81|0.86|0.85|0.88|0.84|0.90|0.85|0.86|
> |Moderate Noise|Qwen3-32B|0.82|0.78|0.87|0.84|0.78|0.71|0.83|0.75|0.85|0.82|0.83|0.79|**0.78**|
> ||Qwen3-8B|0.80|0.75|0.86|0.82|0.76|0.69|0.81|0.71|0.82|0.79|0.80|0.76|0.76|
> ||Qwen3-4B|0.74|0.70|0.83|0.76|0.70|0.62|0.75|0.65|0.77|0.74|0.75|0.71|0.72|
> ||Qwen3-1.7B|0.68|0.62|0.79|0.70|0.63|0.54|0.68|0.57|0.70|0.64|0.69|0.65|0.66|
> ||Llama-3.1-8B|0.77|0.71|0.84|0.79|0.77|0.68|0.80|0.71|0.81|0.77|0.78|0.74|0.75|
> |Heavy Noise|Qwen3-32B|0.72|0.69|0.90|0.82|0.68|0.72|0.78|0.66|0.78|0.75|0.78|0.71|**0.72**|
> ||Qwen3-8B|0.70|0.67|0.89|0.81|0.67|0.69|0.76|0.65|0.76|0.73|0.74|0.68|0.71|
> ||Qwen3-4B|0.65|0.62|0.85|0.75|0.63|0.65|0.70|0.60|0.70|0.67|0.69|0.63|0.68|
> ||Qwen3-1.7B|0.57|0.51|0.81|0.69|0.55|0.52|0.63|0.48|0.62|0.60|0.64|0.59|0.60|
> ||Llama-3.1-8B|0.67|0.60|0.87|0.78|0.66|0.67|0.75|0.63|0.75|0.71|0.73|0.67|0.68|

---

### Author Response · Authors · 2025-11-30
**Rebuttal Summary (3/3)**

(2) Evaluation under Broader Perturbations

We added the evaluation of the QA task under adversarial paraphrasing and format perturbation. RobustFlow maintains approximately 80% stability on adversarial interpretations and approximately 95% stability on format perturbations. **Although RobustFlow is not explicitly trained for these specific adversarial patterns, it achieves strong generalization capabilities.**

| Perturbation            | Method         | QA       |          |          |          | Avg. |
| ----------------------- | -------------- | -------- | -------- | -------- | -------- | ---- |
|                         |                | HotpotQA |          | DROP     |          |      |
|                         |                | Node     | Graph    | Node     | Graph    |      |
| Adversarial Paraphrases | AFlow          | 0.49     | 0.41     | 0.43     | 0.41     | 0.43 |
|                         | ScoreFlow      | 0.29     | 0.21     | 0.26     | 0.23     | 0.25 |
|                         | **RobustFlow** | **0.83** | **0.76** | **0.81** | **0.77** | **0.79** |
| Format Perturbations    | AFlow          | 0.57     | 0.49     | 0.54     | 0.49     | 0.52 |
|                         | ScoreFlow      | 0.78     | 0.67     | 0.68     | 0.61     | 0.68 |
|                         | **RobustFlow** | **0.97** | **0.94** | **0.96** | **0.95** | **0.96** |



(3) Robustness comparison under no-disturbance conditions

We added an explicit evaluation of the base workflows on clean and unperturbed queries. This confirms that our training framework can successfully guide the model to converge to a single canonical workflow structure for a given task semantics.

| Perturbation         | Method         | Code     |          |           |          | MATH     |          |          |          | QA       |          |          |          | Avg.     |
| -------------------- | -------------- | -------- | -------- | --------- | -------- | -------- | -------- | -------- | -------- | -------- | -------- | -------- | -------- | -------- |
|                      |                | MBPP     |          | HumanEval |          | GSM8K    |          | MATH     |          | HotpotQA |          | DROP     |          |          |
|                      |                | Node     | Graph    | Node      | Graph    | Node     | Graph    | Node     | Graph    | Node     | Graph    | Node     | Graph    |          |
| Without Perturbation | AFlow          | 0.68     | 0.62     | 0.71      | 0.65     | 0.58     | 0.52     | 0.55     | 0.48     | 0.64     | 0.59     | 0.62     | 0.57     | 0.60     |
|                      | ScoreFlow      | 0.86     | 0.82     | 0.89      | 0.85     | 0.81     | 0.78     | 0.79     | 0.75     | 0.83     | 0.79     | 0.82     | 0.78     | 0.81     |
|                      | **RobustFlow** | **0.99** | **0.98** | **0.99**  | **0.99** | **0.98** | **0.97** | **0.96** | **0.95** | **0.98** | **0.97** | **0.98** | **0.96** | **0.97** |

(4) Robustness-performance trade-off

RobustFlow employs an adversarial training paradigm, which inherently involves a trade-off between performance and robustness. We provided detailed analysis in Sec. 5.2 and examples of workflows generated by both RobustFlow and ScoreFlow for the same task in appendix A.5, demonstrating why performance may decrease as robustness increases. Compared to the random behavior of baseline models, **the small performance loss is a necessary price to pay for achieving such high reliability**.

(5) Reproducibility and implementation details

All code, configuration files, and hyperparameter settings used in our experiments have been released in the anonymous repository \url{https://anonymous.4open.science/r/RobustFlow-1B19}. We updated Appendix A.4 with concrete implementation details for each baseline, including the codebases and key hyperparameters. We have also carefully revised the manuscript to clarify the expressions that were noted as ambiguous in the reviews, ensuring a more precise presentation of our work.



We fully understand that the system rollback has created additional burden for the new Area Chair. We provide this brief summary only in the hope of presenting as clearly as possible the current state of the paper and the efforts we have made to address the reviewers’ comments. Regardless of the final decision, we sincerely thank you for your time and for reviewing our work amid your busy schedule.

Best regards,

The Authors

---

### Author Response · Authors · 2025-11-30
**Rebuttal Summary (2/3)**

**(2) Major Responses at Rebuttal**

The main concerns raised by the four reviewers can be summarized into the following 5 points:

a) Lack of empirical evidence on generalization across diverse model architectures and sizes (Pnjn, 79uy);

b) Lack of comprehensive evaluation across broader perturbations types (EUP9, 79uy);

c) Lack of evaluation comparison under undisturbed conditions (EUP9);

d) Lack of in-depth analysis regarding the robustness-performance trade-off and robustness mechanisms (Pnjn, Jb4K, EUP9);

e) Lack of sufficient details for reproducibility and implementation specifics (EUP9, 79uy).

**During the discussion phase, we carried out the following categories of improvements and experiments around these concerns.**



(1) Generalization experiments across base models

We added the robustness evaluation across different model families (Qwen3, Llama-3.1) and scales (1.7B, 4B, 8B, 32B). **RobustFlow maintains high robustness even as model size decreases.** This demonstrates that our training framework is model-independent and effective across various architectures.

||AFlow|ScoreFlow|RobustFlow (Qwen3-32B)|RobustFlow (Qwen3-8B)|RobustFlow (Qwen3-4B)|RobustFlow (Qwen3-1.7B)|RobustFlow (Llama-3.1-8B)|
|--------------|-----|---------|---------------------|--------------------|--------------------|----------------------|------------------------|
|Requirement|0.44|0.71|0.82|**0.83**|0.78|0.72|**0.83**|
|Paraphrasing|0.49|0.73|**0.88**|0.86|0.82|0.78|0.87|
|Light Noise|0.42|0.60|**0.89**|0.87|0.82|0.76|0.86|
|Moderate Noise|0.54|0.53|**0.78**|0.76|0.72|0.66|0.75|
|Heavy Noise|0.65|0.49|**0.72**|0.71|0.68|0.60|0.68|


|Perturbation|RobustFlow base model|Code||||MATH||||QA||||Avg.|
|:-------------|:--------------------|:---|:----|:--------|:----|:----|:----|:---|:----|:-------|:----|:---|:----|:-------|
|||MBPP||HumanEval||GSM8K||MATH||HotpotQA||DROP|||
|||Node|Graph|Node|Graph|Node|Graph|Node|Graph|Node|Graph|Node|Graph||
|Requirement|Qwen3-32B|0.89|0.83|0.95|0.93|0.87|0.77|0.84|0.80|0.88|0.83|0.86|0.82|0.82|
||Qwen3-8B|0.87|0.80|0.94|0.90|0.85|0.76|0.81|0.75|0.86|0.81|0.83|0.78|**0.83**|
||Qwen3-4B|0.82|0.76|0.90|0.87|0.78|0.69|0.79|0.68|0.81|0.76|0.76|0.73|0.78|
||Qwen3-1.7B|0.76|0.71|0.88|0.82|0.71|0.60|0.73|0.59|0.74|0.70|0.69|0.65|0.72|
||Llama-3.1-8B|0.84|0.76|0.92|0.88|0.88|0.79|0.83|0.77|0.87|0.82|0.84|0.79|**0.83**|
|Paraphrasing|Qwen3-32B|0.92|0.88|0.98|0.96|0.91|0.88|0.87|0.83|0.93|0.88|0.94|0.85|**0.88**|
||Qwen3-8B|0.91|0.85|0.97|0.93|0.88|0.84|0.85|0.80|0.90|0.85|0.91|0.81|0.86|
||Qwen3-4B|0.87|0.81|0.94|0.90|0.83|0.78|0.81|0.74|0.85|0.80|0.83|0.77|0.82|
||Qwen3-1.7B|0.81|0.75|0.92|0.86|0.76|0.69|0.77|0.68|0.79|0.73|0.76|0.69|0.78|
||Llama-3.1-8B|0.89|0.81|0.96|0.91|0.90|0.86|0.86|0.82|0.91|0.86|0.92|0.82|0.87|
|Light Noise|Qwen3-32B|0.95|0.93|0.96|0.90|0.92|0.85|0.88|0.89|0.91|0.87|0.94|0.89|**0.89**|
||Qwen3-8B|0.93|0.90|0.95|0.88|0.89|0.82|0.85|0.85|0.89|0.83|0.91|0.86|0.87|
||Qwen3-4B|0.89|0.84|0.92|0.83|0.84|0.75|0.80|0.79|0.84|0.78|0.86|0.80|0.82|
||Qwen3-1.7B|0.83|0.78|0.89|0.76|0.75|0.66|0.74|0.70|0.76|0.71|0.79|0.73|0.76|
||Llama-3.1-8B|0.90|0.85|0.93|0.85|0.90|0.81|0.86|0.85|0.88|0.84|0.90|0.85|0.86|
|Moderate Noise|Qwen3-32B|0.82|0.78|0.87|0.84|0.78|0.71|0.83|0.75|0.85|0.82|0.83|0.79|**0.78**|
||Qwen3-8B|0.80|0.75|0.86|0.82|0.76|0.69|0.81|0.71|0.82|0.79|0.80|0.76|0.76|
||Qwen3-4B|0.74|0.70|0.83|0.76|0.70|0.62|0.75|0.65|0.77|0.74|0.75|0.71|0.72|
||Qwen3-1.7B|0.68|0.62|0.79|0.70|0.63|0.54|0.68|0.57|0.70|0.64|0.69|0.65|0.66|
||Llama-3.1-8B|0.77|0.71|0.84|0.79|0.77|0.68|0.80|0.71|0.81|0.77|0.78|0.74|0.75|
|Heavy Noise|Qwen3-32B|0.72|0.69|0.90|0.82|0.68|0.72|0.78|0.66|0.78|0.75|0.78|0.71|**0.72**|
||Qwen3-8B|0.70|0.67|0.89|0.81|0.67|0.69|0.76|0.65|0.76|0.73|0.74|0.68|0.71|
||Qwen3-4B|0.65|0.62|0.85|0.75|0.63|0.65|0.70|0.60|0.70|0.67|0.69|0.63|0.68|
||Qwen3-1.7B|0.57|0.51|0.81|0.69|0.55|0.52|0.63|0.48|0.62|0.60|0.64|0.59|0.60|
||Llama-3.1-8B|0.67|0.60|0.87|0.78|0.66|0.67|0.75|0.63|0.75|0.71|0.73|0.67|0.68|

---

### Author Response · Authors · 2025-11-30
**Rebuttal Summary (1/3)**

Dear Area Chair,

Thank you for taking the time to review our submission under these special circumstances. We would like to briefly and concisely clarify two points:

(1) **how the original reviewers adjusted their scores during the discussion phase based on our rebuttal and new experiments, before the leak incident occurred**;

(2) **the additional experiments and revisions we have completed in direct response to the reviewers’ core concerns**, so that you can more efficiently understand the current status of the paper.

During the discussion phase, the score adjustment from Reviewer EUP9 occurred on November 27, 3:23 AM, prior to the leak incident. Crucially, this update was a direct response to our rebuttal and new experiments posted on Nov 26, 2:18 AM, where we addressed the reviewer's specific concerns. The score changes and current status for the four reviewers are as follows:

| Reviewer | Original | After Discussion |
| -------- | -------- | ---------------- |
| Pnjn     | 8        | 8 (Not Reply)    |
| EUP9     | 2        | 4 (Nov 27, 3:23) |
| Jb4K     | 4        | 4 (Not Reply)    |
| 79uy     | 4        | 4 (Not Reply)    |



**(1) Discussion phase before the leakage**

**Reviewer Pnjn**: score 8, as the reviewer did not respond during the discussion phase. The reviewer raised insightful questions regarding scalability across model architectures and sizes, failure modes mechanisms, and potential functional metrics. We adopted their suggestions by extending experiments to Llama-3.1 (8B) and the Qwen3 family (32B/8B/4B/1.7B) to demonstrate broader applicability, and expanding a detailed discussion on trade-offs and failure mode analysis (Sec. 5.2).

**Reviewer EUP9**: raised the score from 2 to 4 on Nov 27, 3:23 AM, after we added evaluation of original workflow quality on Nov 26, 2:18 AM. The reviewer's main concerns focused on methodological clarity, scope of robustness definition, and experimental configurations. We addressed these directly by clarifying the method definitions (Sec. 4.2), demonstrating the fairness of baseline comparisons (Appendix A.4), and providing full implementation details (\url{ https://anonymous.4open.science/r/RobustFlow-1B19}). The score increase reflects the reviewer's acknowledgement of these technical improvements.

**Reviewer Jb4K**: score 4, as the reviewer did not respond during the discussion phase. The reviewer's main concerns focused on the robustness-performance trade-off, the validity of the canonical strategy, and the definition of execution scores. We addressed these comprehensively by formalizing the execution score definition (Sec. 4.2), adding a dedicated discussion justifying the trade-off (Sec. 5.2), and providing extended case studies to demonstrate the method's validity (Appendix A.5, A.6).

**Reviewer 79uy**: score 4, as the reviewer did not respond during the discussion phase. The reviewer's main concerns focused on model generalization across architectures and robustness against other perturbations. We addressed these comprehensively by extending experiments to Llama-3.1 (8B) and the Qwen3 family (32B/8B/4B/1.7B) to demonstrate broader applicability, validating performance under adversarial perturbations, and revising figures to improve readability as requested.

---

### Meta-Review · Area_Chair_rHJm · 2025-12-24

**Summary:**

The summary of the reviewers' concerns that informed my suggested decision for this paper, 1) The setup of workflow generation robustness and the task scenarios are overly simplistic and there is a qualitative difference between prompt-based exploration optimization and parameter-based training optimization. Unfair comparisons still persist after the authors give the rebuttal; 2) The evaluation datasets mainly consist of relatively simple QA-style or reasoning tasks, which can often be solved by a single LLM without requiring genuine multi-step agentic planning or tool interaction. The current evaluation does not demonstrate that RobustFlow generalizes beyond easy tasks; 3) Since agentic workflow systems are intended precisely for such non-trivial, multi-stage problems, the current evaluation does not demonstrate that RobustFlow generalizes beyond easy tasks.

**Reviewer Concerns:**

These concerns raised by Reviewer Pnjn and  Reviewer Jb4K are addressed by the reviewer. Even Reviewer Jb4K given the rating as 4 and did not give the response after the rebuttal. The authors given more detailed discussions on the point, i.e., Forcing the model toward a single canonical workflow might prevent it from finding a more optimal, specialized workflow that could be better suited for a specific phrasing of an instruction. The other related explanations are also presented by the authors, i.e., give more clear definition of the `execution score' in ScPO.

The concerns raised by Reviewer EUP9 and Reviewer 79uy are still outstanding, especially the following comments raised by Reviewer EUP9. The setup of workflow generation robustness and the task scenarios are overly simplistic and there is a qualitative difference between prompt-based exploration optimization and parameter-based training optimization. Unfair comparisons still persist after the authors give the rebuttal. The following concern is also still outstanding. Since agentic workflow systems are intended precisely for such non-trivial, multi-stage problems, the current evaluation does not demonstrate that RobustFlow generalizes beyond easy tasks.

**Reviewer Scores:**

I think Reviewer Pnjn will keep the original rating (rating: 8) since the raised concerns are addressed by the reviewer. Reviewer Jb4K will raise the the original rating (rating: 4) to positive rating since the authors well given the related explanations raised by this reviewer.

Reviewer EUP9 raised the rating from 2 to 4 before the leakage and I think this reviewer will keep this rating since some important concerns are still outstanding, i.e, The setup of workflow generation robustness and the task scenarios are overly simplistic and there is a qualitative difference between prompt-based exploration optimization and parameter-based training optimization. Unfair comparisons still persist after the authors give the rebuttal as reviewer mentioned. Likewise, I think Reviewer 79uy will keep the original rating considering some concerns are still outstanding, i.e., the current evaluation does not demonstrate that RobustFlow generalizes beyond easy tasks.

---

### Decision · Program_Chairs · 2026-01-26

Reject